# IL-22-dependent dysbiosis and mononuclear phagocyte depletion contribute to steroid-resistant gut graft-versus-host disease in mice

Qingxiao Song [1,2,3], Xiaoning Wang [1,2,4], Xiwei Wu[5], Tae Hyuk Kang [5], Hanjun Qin[5], Dongchang Zhao[6], Robert R. Jenq[7], Marcel R. M. van den Brink [8], Arthur D. Riggs[1], Paul J. Martin [9] & Yuan-Zhong Chen [3 ✉] & Defu Zeng [1,2 ✉]

Efforts to improve the prognosis of steroid-resistant gut acute graft-versus-host-disease (SR-Gut-aGVHD) have suffered from poor understanding of its pathogenesis. Here we show that the pathogenesis of SR-Gut-aGVHD is associated with reduction of IFN-$\gamma^+$ Th/Tc1 cells and preferential expansion of IL-17$^-$IL-22$^+$ Th/Tc22 cells. The IL-22 from Th/Tc22 cells causes dysbiosis in a Reg3γ-dependent manner. Transplantation of IFN-γ-deficient donor CD8$^+$ T cells in the absence of CD4$^+$ T cells produces a phenocopy of SR-Gut-aGVHD. IFN-γ deficiency in donor CD8$^+$ T cells also leads to a PD-1-dependent depletion of intestinal protective CX3CR1$^{hi}$ mononuclear phagocytes (MNP), which also augments expansion of Tc22 cells. Supporting the dual regulation, simultaneous dysbiosis induction and depletion of CX3CR1$^{hi}$ MNP results in full-blown Gut-aGVHD. Our results thus provide insights into SR-Gut-aGVHD pathogenesis and suggest the potential efficacy of IL-22 antagonists and IFN-γ agonists in SR-Gut-aGVHD therapy.

[1] Diabetes and Metabolism Research Institute, The Beckman Research Institute of City of Hope, Duarte, CA, USA. [2] Hematologic Malignancies and Stem Cell Transplantation Institute, The Beckman Research Institute of City of Hope, Duarte, CA, USA. [3] Fujian Medical University Center of Translational Hematology, Fujian Institute of Hematology, and Fujian Medical University Union Hospital, Fuzhou, China. [4] Department of Hematology, The First Affiliated Hospital of Xi'an Jiaotong University, Xi'an, Shaanxi, China. [5] Department of Integrative Genomics Core, The Beckman Research Institute of City of Hope, Duarte, CA, USA. [6] The Tisch Cancer Institute and Division of Hematology/Medical Oncology, The Icahn School of Medicine at Mount Sinai Hospital, New York, NY, USA. [7] Departments of Genomic Medicine and Stem Cell Transplantation Cellular Therapy, University of Texas MD Anderson Cancer Center, Houston, TX, USA. [8] Department of Medicine, Adult Bone Marrow Transplant Service, Memorial Sloan Kettering Cancer Center, New York, NY, USA. [9] Fred Hutchinson Cancer Research Center, University of Washington, Seattle, WA, USA. ✉email: chenyz@fjmu.edu.cn; dzeng@coh.org

Acute graft-versus-host disease (aGVHD) is mediated by alloreactive donor CD4[+] and CD8[+] T cells after allogeneic hematopoietic cell transplantation (HCT)[1,2]. The gastrointestinal tract is a prominent target of aGVHD, and the severity of damage in the intestine (gut) determines the outcome of aGVHD[3]. IFN-γ[+] Th1/Tc1 cells play the dominant role in damaging intestinal Paneth cells that produce Reg3γ, a protein that has bactericidal activity against Gram-positive bacteria[4] and maintains the homeostasis of intestinal microbiome[5]. aGVHD damage of intestinal epithelial cells and Paneth cells results in dysbiosis that exacerbates Gut-aGVHD[6–11]. Under inflammatory conditions, enhanced production of IL-22 and Reg3γ in gut tissues can also induce dysbiosis and pathogen colonization[12]. IL-22 in gut tissues can be produced by innate lymphocytes, NK and NKT cells, as well as Th17 and Th22 cells[13,14]. Th/Tc17 cells include IL-17A[+]IL-22[−]and IL-17A[+]IL-22[+] subsets, and their differentiation is regulated by RORα and RORc (RORγt)[15,16]. Th/Tc22 differentiation is regulated by AHR, and in Th22 cells, AHR expression is augmented by RORγt and suppressed by T-bet[17]. Th/Tc22 cells are IL-22[+]IL-17A[−17], and human Th22 cells can be IL-22[hi]IFN-γ[lo]IL-17A[−18].

In healthy individuals, CX3CR1[+] intestinal mononuclear phagocytes (MNP) play an important role in clearing enteroinvasive pathogens and preventing pathogen translocation from the intestinal lumen into mesenteric lymph nodes (MLN) and the liver[19–21]. CX3CR1[+] MNP also promote epithelial barrier repair[22] and regulate Th17 and Treg differentiation[23,24]. Recipient-derived hematopoietic cells including CX3CR1[+] MNP are targets of aGVHD and are replaced with donor-derived cells[25]. How donor-derived CX3CR1[+] MNP cells regulate Gut-aGVHD is unknown.

Corticosteroids are used for initial treatment of aGVHD[26]. Some patients develop steroid-resistant or refractory (SR) Gut-aGVHD, and the pathogenesis of SR-Gut-aGVHD remains enigmatic[27,28]. Steroid-treatment effectively suppresses Th1/Tc1 but not Th17[29], and IL-17A[+]CD4[+] T cells infiltrate the intestinal tissues of patients with SR-Gut-aGVHD[30]. However, targeting steroid-resistant T cells with ATG or anti-CD25 in patients with SR-Gut-aGVHD has not been effective[27]. A recent study with murine models indicated that T cells were dispensable for SR-GVHD pathogenesis[31]. Other studies indicate that while IL-22 from host-type innate lymphocytes augment intestinal epithelial stem cell survival and reduce Gut-aGVHD[32,33], IL-22 from donor T cells augments Gut-aGVHD[34,35], although its mechanisms and cellular source (Th17 versus Th22) remains unclear.

In the current studies, considering the differential effect of steroid on T-cell subsets, the dual effect of IL-22 from Th/Tc17 or Th/Tc22 on regulating intestinal microbiome profiles, and the complex function of CX3CR1[hi] MNP in regulating T-cell differentiation and bacterial translocation in the gut tissues, we explore the roles of Th/Tc17 and Th/Tc22 as well as CX3CR1[hi] MNP cells in the pathogenesis of SR-Gut-aGVHD. We observe that steroid treatment reduces IFN-γ[+] Th1/Tc1 but preferentially augments expansion of Th/Tc22 cells; IL-22 from Th/Tc22 cells induces dysbiosis; lack of IFN-γ leads to depletion of protective donor-type CX3CR1[hi] MNP cells in the gut tissues; simultaneous induction of dysbiosis by IL-22 and depletion of CX3CR1[hi] MNP cells result in full-brown SR-Gut-aGVHD. Our studies provide insights into pathogenesis of SR-Gut-aGVHD and open an avenue towards developing approaches for preventing and treating SR-Gut-aGVHD.

## Results
### Establish a murine model of steroid-resistant acute gut GVHD.
Owing to lacking a good murine model of SR-Gut-aGVHD, we

attempted to establish a new murine model. Lethal TBI-conditioned BALB/c mice (H-2[d]) were given spleen (SPL) cells containing $1.5 \times 10^6$ T cells together with T cell-depleted bone marrow (TCD-BM) cells ($2.5 \times 10^6$) from major histocompatibility complex (MHC)-mismatched C57BL/6 (H-2[b]) donors. Three days after HCT, recipients were given a single injection of dexamethasone (DEX) at 5 mg/Kg. This treatment ameliorated Gut-aGVHD as indicated by prevention of diarrhea, reduction of bodyweight loss, and enabling recipients to survive for up to 20 days after HCT with mild clinical GVHD. In contrast, controls treated with saline all (12/12) died by day 11 with diarrhea and severe bodyweight loss (Fig. 1a). After day 20, however, the DEX-treated mice developed bodyweight loss and began to die. Additional DEX administration on days 10, 15, and 20 (total 4-DEX) did not prevent this deterioration or improve survival as compared with single DEX treatment on day 3 (Fig. 1a). High serum concentration of Reg3α, ST2 and sTNFR has been associated with SR-Gut-aGVHD in patients[27]. The 4-DEX-treated recipients also showed marked increase in serum Reg3γ, ST2, and sTNFR as compared with non-GVHD or 1-DEX-treated GVHD recipients (Fig. 1b). At 30 days after HCT, 4-DEX-treated recipients had higher numbers of Paneth cells in the small intestine but more severe GVHD in the colon compared with 1-DEX-treated recipients (Fig. 1c, d). These results indicate that Gut-aGVHD in recipients given 4-DEX treatment has evolved to SR-Gut-aGVHD.

### SR-Gut-aGVHD is associated with expansion of AHR[+] Th/Tc22 cells.
As steroids augment Th/Tc1 apoptosis[29] and reduce tissue release of TGF-β but not IL-6[36,37] and as TGF-β and IL-6 reciprocally regulate differentiation of Th17 and Th22[38–40], we analyzed the changes of Th/Tc1, Th/Tc17, and Th/Tc22 subsets in recipients with or without DEX treatment. Because the saline-treated control mice all died ~10 days after HCT, we first compared the T-cell subsets in the spleen (SPL), MLN, and colon tissues at 7 days after HCT. As compared with saline controls, DEX treatment on day 3 significantly reduced the percentages of IFN-γ[+] Th/Tc1 cells in SPL, MLN, and colon tissues (Fig. 1e). At the same time, the numbers of Th/Tc17 cells were very low, even in the recipients treated with DEX, there were only 1–5% IL-17A[+]IL-22[−]CD4[+] Th17 cells and IL-17A[+]IL-22[−]CD8[+] Tc17 cells among CD4[+] and CD8[+] T cells (Fig. 1f). The percentages of IL-17A[+]IL-22[+] CD4[+] T cells were <0.5%, and IL-17A[+]IL-22[+] CD8[+] T cells were essentially undetectable. The percentage of IL-17A[−]IL-22[+]CD4[+] was ~0.5–1% and IL-17A[−]IL-22[+]CD8[+] T cells were barely detectable (Fig. 1f). These results indicate that DEX treatment early after HCT mainly reduces Th/Tc1 expansion, with little impact on Th/Tc17 or Th/Tc22 cells.

To analyze the T-cell subsets in SR-Gut-aGVHD, recipients were transplanted with spleen cells containing $0.75 \times 10^6$ instead of $1.5 \times 10^6$ T cells to allow the saline control group to survive for more than 25 days after HCT. On day 25, as compared with saline treatment, 1-DEX and 4-DEX treatments both decreased the numbers of donor-type IL-17A[+]IL-22[−]CD4[+] and CD8[+] T cells in the MLN, although the reduction in the spleen was variable, and no significant difference between 1-DEX and 4-DEX treatment (Fig. 2a). DEX treatment did not change the numbers of IL-17A[+]IL-22[+] CD4[+] or CD8[+] T cells in the spleen or MLN (Fig. 2b). Interestingly, 4-DEX treatment significantly increased both the percentages and the yields of IL-17A[−]IL-22[+] CD4[+] and CD8[+] T cells, in both the spleen and MLN, as compared with 1-DEX treatment or saline (Fig. 2c).

IL-17A[+]IL-22[−]CD4[+] T cells were RORγt[+]AHR[−] Th17 cells, whereas only small portion of IL-17A[+]IL-22[−]CD8[+] T cells were RORγt[+]AHR[−] Tc17 cells; in contrast, IL-17A[−]IL-22[+] CD4[+] and

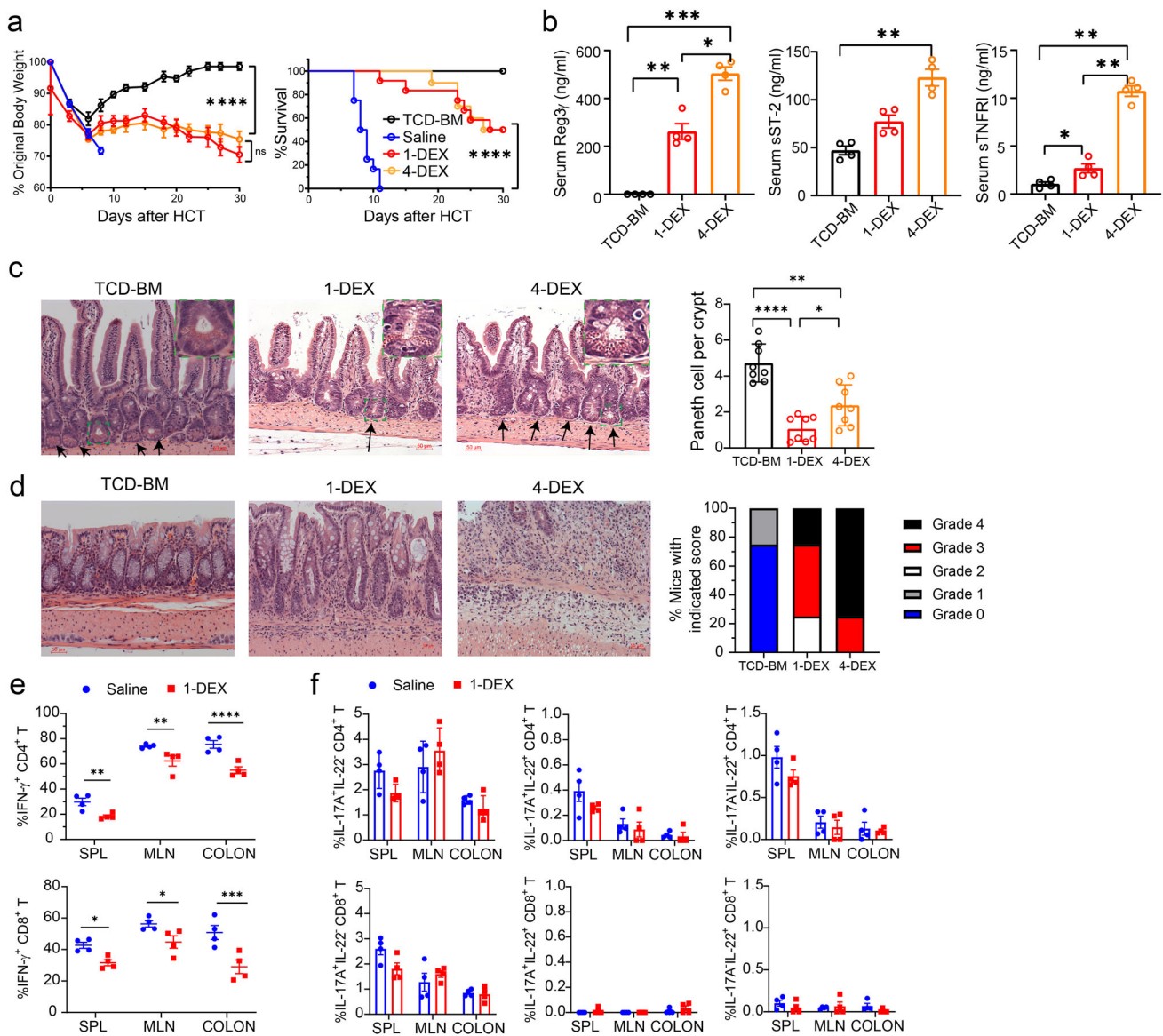

**Fig. 1 Establishing a murine model of steroid-resistant acute gut GVHD (SR-Gut-aGVHD).** Lethally irradiated WT BALB/c recipients were engrafted with splenocytes containing $1.5 \times 10^6$ T cells together with T cell depleted bone marrow (TCD-BM) ($2.5 \times 10^6$) from WT C57BL/6 donors. Recipients were given a single iv injection of dexamethasone (DEX, 5 mg/kg) on day 3 (1-DEX) or four total injections on days 3, 10, 15, and 20 (4-DEX) after HCT, with saline used as control. **a** Plots of means ± SEM of %Original bodyweight of each individual mice at each time point and recessive curve of %survival. $n = 8$ (TCD-BM), 10 (4-DEX), 12 (Saline & 1-DEX) from two replicate experiments. **b** Recipient serum concentrations of Reg3γ, ST2, and sTNFRI were measured on day 25 after HCT. Means ± SEM are shown, $n = 4$ combined from two replicate experiments. **c, d** Histopathology of small intestine and colon was evaluated day 25 after HCT. Representative micrographic photos of small intestine (original magnification, ×200), and means ± SEM of the numbers of Paneth cell per crypt are showed in **c**. Representative micrographic photos of colon (original magnification, ×200) and % mice with indicated histopathological scores are shown in **d**. $n = 8$ biologically independent mice per group combined from two replicate experiments. **e, f** Cytokine profile of T cells from SPL (spleen), MLN (mesenteric lymph node), and colon were measured on day 7 after HCT. Means ± SEM of %IFN-γ+ cells among CD4+ and CD8+ T-cell subsets are shown in **e**, and means ± SEM of %IL-17A+IL-22−, IL-17A+IL-22+ and IL-17A−IL-22+ cells among CD4+ and CD8+ T-cell subsets are shown in **f**. $n = 4$ combined from two experiments. Each dot represents one mouse. Nonlinear regression (curve fit) was used and a two-tailed $p$ value was calculated for bodyweight comparisons. Log-rank test was performed with two-tailed $p$ value for survival comparisons. One-way ANOVA with Tukey's multiple comparisons test, with the Greenhouse-Geisser correction was used for the comparisons in **b, c**; unpaired two-tailed Student's $t$ test corrected for multiple comparisons using the Holm–Sidak method was used to compare means in **e**. ****$p < 0.0001$; **b**, Reg3γ: *$p = 0.0133$, **$p = 0.0086$, ***$p = 0.0008$; ST2: **$p = 0.0073$(ST2); sTNFRI: *$p = 0.0208$, **$p = 0.0020$(TCD-BM vs 4-DEX), **$p = 0.0077$ (1-DEX vs 4-DEX); **c** *$p = 0.0371$ (TCD vs 4-DEX), **$p = 0.0011$ (TCD vs 1-DEX), **$p = 0.0023$ (1-DEX vs 4-DEX); **e** IFN-γ+CD4+ T: **$p = 0.0080$ (SPL), **$p = 0.0069$ (MLN), ****$p < 0.0001$; IFN-γ+CD8+ T: *$p = 0.0291$ (SPL), *$p = 0.0239$ (MLN), ***$p = 0.00019$.

CD8+ T cells were both AHR+RORγt− Th/Tc22 cells (Fig. 2d). Consistent with day 7 results, 4-DEX treatment significantly reduced the percentage of IFN-γ+ Th/Tc1 cells in gut tissues (Fig. 2e). We also tested three healthy human PBMC. PBMC from one

of these donors induced SR-Gut-aGVHD with 4-DEX treatment. The Xeno-SR-Gut-aGVHD induced by these cells was also associated with expansion of IL-17A−IL-22+ CD4+ and CD8+ T cells in the colon tissue (Fig. S1). These results indicate that

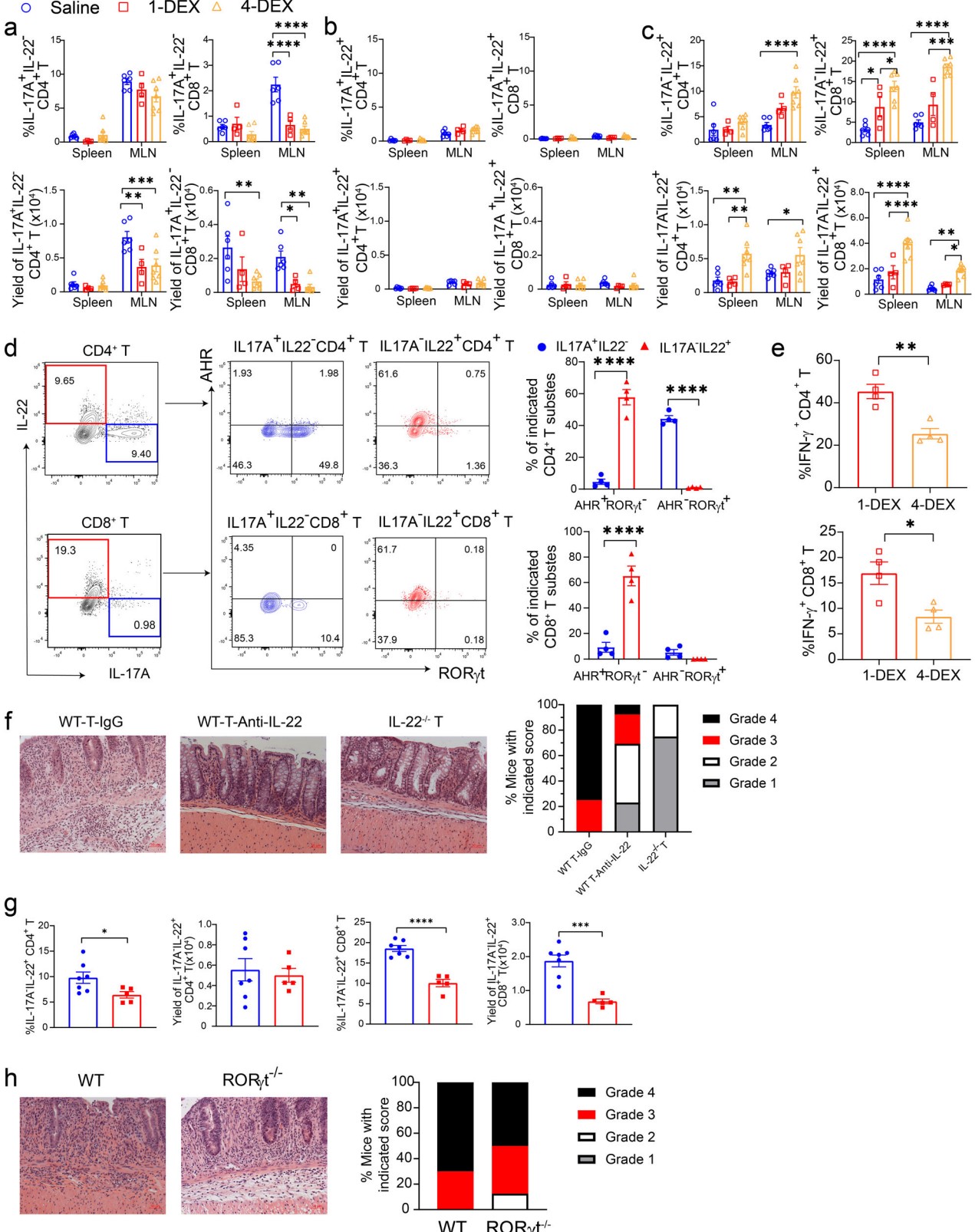

prolonged steroid treatment augments expansion of AHR[+]RORγt[−]IL-17A[−]IL-22[+] CD4[+] and CD8[+] Th/Tc22 cells while reducing RORγt[+]AHR[−]IL-17A[+]IL-22[−] Th/Tc17 cells; these results also indicate that SR-Gut-aGVHD pathogenesis is associated with preferential expansion of AHR[+] Th/Tc22 cells in the colon.

**IL-22 from Th/Tc22 cells is required for induction of SR-Gut-aGVHD.** Because we observed preferential expansion of Th/Tc22 cells in SR-Gut-aGVHD recipients (Fig. 2), we tested whether IL-22 from Th/Tc22 but not from Th/Tc17 cells caused SR-Gut-aGVHD. Accordingly, BALB/c recipients engrafted with $1.5 \times 10^6$ splenic T cells and TCD-BM cells from WT C57BL/6 donors were

**Fig. 2 IL-22 from Th/Tc22 cells are required for induction of SR-Gut-aGVHD. a–c** Recipients were given 1-DEX, 4-DEX, or saline treatment as described in Fig. 1. Cytokine profiles of T cells from spleen and MLN were measured on day 25. Means ± SEM of % and yield of IL17A$^+$IL-22$^-$, IL17A$^+$IL-22$^+$, and IL17A$^-$IL-22$^+$ among CD4$^+$ and CD8$^+$ T subsets are shown in **a–c**, respectively. Each dot represents one mouse. $n = 4$ (1-DEX), 6 (Saline), 7 (4-DEX) combined from two replicates. **d** Representative flow cytometry patterns and means ± SEM of AHR$^+$RORγt$^-$ and AHR$^-$RORγt$^+$ among IL-17A$^+$IL-22$^-$ or IL-17A$^-$IL-22$^+$ CD4$^+$ and CD8$^+$ subsets. Each dot represents one mouse, $n = 4$, two replicates. **e** Means ± SEM of % and yield of IFN-γ$^+$CD4$^+$ and IFN-γ$^+$CD8$^+$ T in ileum. Each dot represents one mouse, $n = 4$, two replicates. **f** Recipients engrafted with splenic T from WT or IL-22$^{-/-}$ donors combined with TCD-BM from WT donors were given 4-DEX treatment. The recipients of WT-T were given additional treatment of anti-IL-22 mAb or IgG at 150 μg every 3 days from days 12 to 21 after HCT. On day 25, histopathology of colon was evaluated. Representative micrographs (200×) and %mice with indicated histopathological scores are shown; $n = 4$ (WT-IgG & IL-22$^{-/-}$ T), 12 (WT-anti-IL-22) biologically independent mice combined from two replicates. **g, h** Recipients engrafted with T cells from WT or T-RORγt$^{-/-}$ combined with TCD-BM from WT donors were evaluated for cytokine profile of MLN T cells and histopathology of colon on day 25. **g** Means ± SEM of % and yields of IL17A$^-$IL-22$^+$ cells among CD4$^+$ and CD8$^+$ T-cell subsets; each dot represents one mouse, $n = 5$ (RORγt$^{-/-}$ T), 7 (WT T) combined from two replicates. **h** Representative micrographs of colon (200×) and %mice with indicated histopathological scores; $n = 8$ biologically independent mice combined from two replicates. $P$ value was determined by two-way ANOVA with Tukey's (**a–c**) or Sidak's (**d**), unpaired two-tailed Student' $t$ test (**e, g**). **a** ****$p < 0.0001$; yield of IL17A$^+$IL-22$^-$CD4$^+$: **$p = 0.0013$, ***$p = 0.0004$; yield of IL17A$^+$IL-22$^-$CD8$^+$: *$p = 0.0416$, **$p = 0.0024$ (spleen, saline vs 4-DEX),**$p = 0.0074$ (MLN, Saline vs 4-DEX); **c** ****$p < 0.0001$; yield of IL17A$^-$IL-22$^+$CD4$^+$: **$p = 0.0024$ (saline vs 4-DEX) **$p = 0.0050$ (1-DEX vs 4-DEX); %IL17A$^-$IL-22$^+$CD8$^+$: *$p = 0.0256$ (saline vs 1-DEX), *$p = 0.0362$ (1-DEX vs 4-DEX); ***$p = 0.001$,****$p < 0.0001$. Yield of IL17A$^-$IL-22$^+$CD8$^+$: ****$p < 0.0001$, **$p = 0.0040$, *$p = 0.0493$. **d** ****$p < 0.0001$. **e** *$p = 0.0155$, **$p = 0.0032$. **g** *$p = 0.0410$, ***$p = 0.0003$, ****$p < 0.0001$. Dexamethasone (DEX), MLN (mesenteric lymph node).

treated with 4-DEX. The recipients were treated with anti-IL-22 or control mouse-IgG1 at a dose of 150 μg every 3 days, from day 12 to day 21 after HCT. To further validate the role of IL-22 from donor-type T cells, in the same experiment, a group of BALB/c recipients were engrafted with $1.5 \times 10^6$ splenic T cells from IL-22$^{-/-}$ C57BL/6 donors with TCD-BM from WT C57BL/6 donors and were treated with 4-DEX. At day 25 after HCT, recipients were evaluated of Gut-aGVHD. Neutralizing IL-22 with anti-IL-22 mAb and preventing IL-22 production by using IL-22$^{-/-}$ T cells both markedly reduced the severity of Gut-aGVHD on day 25 after HCT when compared with the positive control (Fig. 2f). These results indicate that IL-22 from donor-type T cells has a critical role in the pathogenesis of SR-Gut-aGVHD.

The 4-DEX treatment expanded only IL-17A$^-$IL-22$^+$ Th/Tc22 cells, whereas IL-17A$^+$IL-22$^+$ CD4$^+$ and CD8$^+$ Th/Tc17 cells were hardly detectable (Fig. 2b–d). To determine whether IL-22 from RORγt$^+$ Th/Tc17 cells is required for induction of SR-Gut-aGVHD, we compared 4-DEX-treated GVHD recipients given $1.5 \times 10^6$ splenic T cells from WT or RORγt$^{-/-}$ C57BL/6 donors with TCD-BM from WT C57BL/6 donors. RORγt deficiency in donor T cells reduced the percentage but not numbers of IL-17A$^-$IL-22$^+$ Th22 cells in the MLN and reduced both the percentage and yield of IL-17A$^-$IL-22$^+$ Tc22 cells in the MLN on day 25 after HCT (Fig. 2g). The reduction of Th/Tc22 cells with RORγt$^{-/-}$ T cells is consistent with previous reports that RORγt augments AHR$^+$ Th22 differentiation[17]. RORγt deficiency in donor T cells, however, did not significantly change the severity of SR-Gut-aGVHD (Fig. 2h). These results indicate that IL-22 from Th/Tc17 cells is not required for SR-Gut-aGVHD.

**IL-22 from Th/Tc22 cells causes dysbiosis and bacteria translocation in SR-Gut-aGVHD recipients.** Under inflammatory conditions, IL-22 augments pathogen colonization and dysbiosis in gut tissues[12]. We tested whether SR-Gut-aGVHD mediated by IL-22 from Th/Tc22 was associated with dysbiosis. Accordingly, day 25 after HCT, feces from the ileum of 1-DEX-treated Gut-aGVHD recipients and 4-DEX-treated SR-Gut-aGVHD recipients given WT-T cells, 4-DEX-treated non-SR-Gut-aGVHD recipients given IL-22$^{-/-}$ T cells, and non-GVHD recipients given TCD-BM alone were analyzed for 16 S ribosomal RNA sequences. As compared with 1-DEX-treated Gut-aGVHD recipients, 4-DEX-treated SR-Gut-aGVHD recipients showed significant loss of bacterial diversity, and IL-22-deficiency in donor T cells restored bacterial diversity, as judged by the numbers of

species, Shannon index and Fisher index (Fig. 3a). As compared with non-GVHD recipients, 1-DEX-treated Gut-aGVHD recipients showed slight but not significant reduction in bacterial diversity (Fig. 3a). Principal coordinate analysis also showed that bacteria from 4-DEX-treated SR-Gut-aGVHD were distinguishable from those of 1-DEX-treated Gut-aGVHD recipients or non-GVHD recipient, and IL-22-deficiency in donor T cells reduced the difference (Fig. 3b).

Further analyzing bacterial subpopulations at species level showed that 4-DEX-treated SR-gut-aGVHD recipients had marked expansion of Enterococcus_sp._FDAARGOS_553 and that IL-22 deficiency in donor T cells prevented this effect; although the differences in *Lactobacillus murinus* and *E. coli* were not statistically significant (Fig. 3c, d). As a further measure of dysbiosis, liver tissue suspension was cultured for bacteria colony formation. As compared with 1-DEX-treated Gut-aGVHD or non-GVHD recipients, 4-DEX-treated SR-Gut-aGVHD recipients had a marked increase in bacterial colony formation, and IL-22 deficiency in donor T cells prevented this effect (Fig. 3e). These results indicate that donor-type Th/Tc22-mediated SR-Gut-aGVHD is associated with dysbiosis and enhanced bacterial translocation into the liver; and IL-22 deficiency in donor T cells prevents the induction of SR-Gut-aGVHD.

**Gut-aGVHD induced by IFN-γ$^{-/-}$ donor CD8$^+$ T cells is associated with expansion of Tc17 and Tc22 cells.** Although splenic T cells from WT and IFN-γ$^{-/-}$ C57BL/6 donors both induced severe Gut-aGVHD, IFN-γ$^{-/-}$ T cell had no effect on small intestine Paneth cells (Fig. S2). We reported that WT donor CD8$^+$ T cells did not cause Gut-aGVHD in the absence of donor CD4$^+$ T cells;[41] others reported that IFN-γ was required for preventing GVHD mediated by CD8$^+$ T cells[42]. We recently observed that IFN-γ$^{-/-}$ donor CD8$^+$ T cells alone induced severe Gut-aGVHD (see below). As SR-Gut-aGVHD mediated by expansion of IL-22$^+$ Th/Tc22 cells was associated with reduction of IFN-γ$^+$ Th/Tc1 (Figs. 1 and 2), we hypothesized that Gut-aGVHD induced by IFN-γ$^{-/-}$CD8$^+$ T cells could reflect the pathogenesis in SR-Gut-aGVHD.

Accordingly, lethal TBI-conditioned BALB/c recipients were engrafted with spleen cells containing $1.5 \times 10^6$ T cells and BM cells ($2.5 \times 10^6$) from WT or IFN-γ$^{-/-}$ C57BL/6 donors, and the recipients were given a single injection of anti-CD4 mAb to deplete the CD4$^+$ T cells[41]. Under these conditions, recipients given IFN-γ$^{-/-}$CD8$^+$ T cells developed aGVHD, but recipients given WT CD8$^+$ T cells did not (Fig. 4a). IFN-γ$^{-/-}$CD8$^+$ T cells

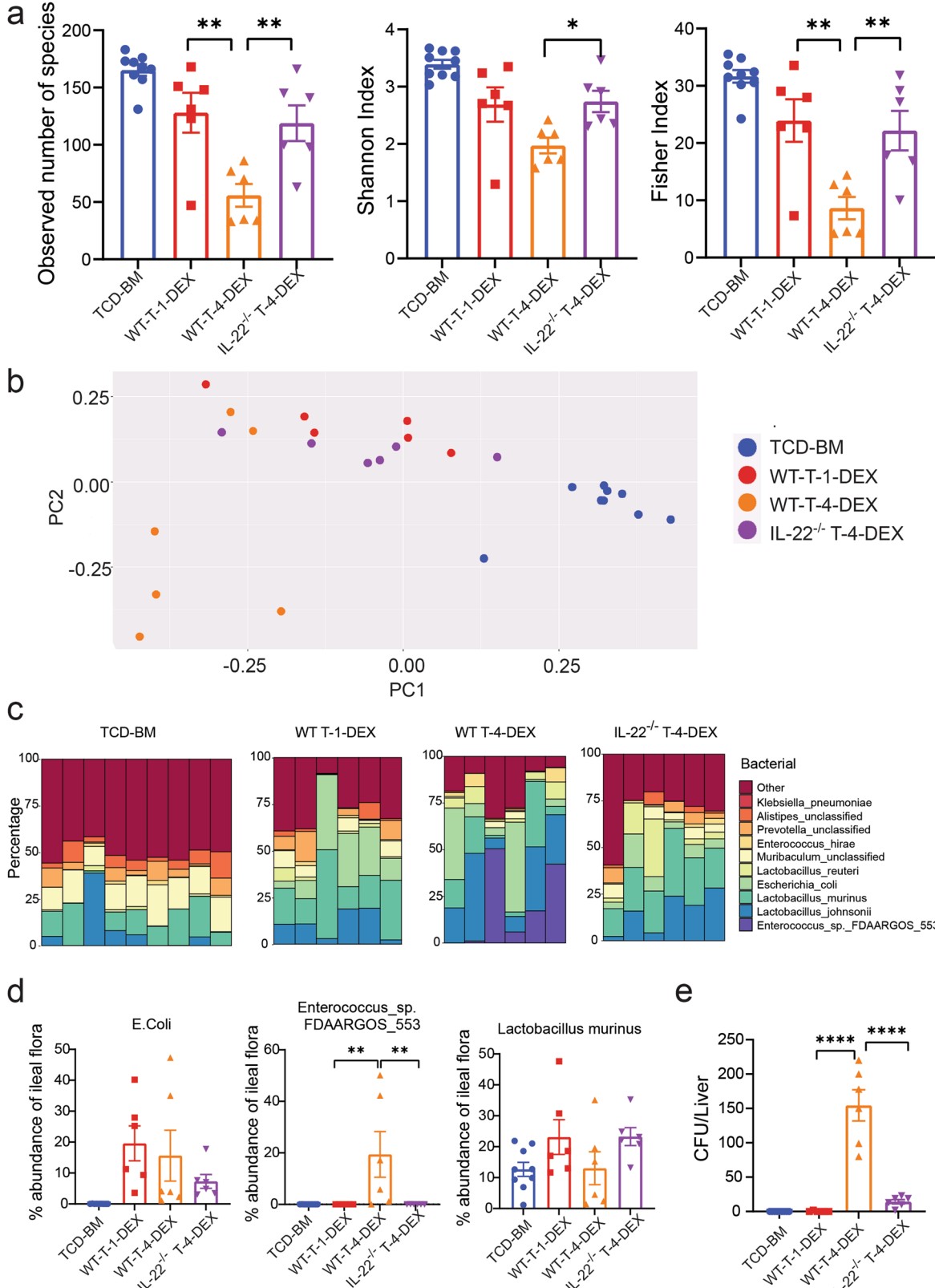

did not induce disease in syngeneic or MHC I-matched recipients (Fig. 4a). Recipients given IFN-$\gamma^{-/-}$-SPL cells developed severe infiltration in the colonic submucosa from days 7 to14 after HCT (Fig. 4b), with little damage in the epithelial cells or Paneth cells in the ileum (Fig. S3a, b). Expression of Defensin-α1 and Defensin-α3 mRNA in the ileal tissue of the recipients was higher

in recipients given IFN-$\gamma^{-/-}$-SPL cells than in those given WT-SPL cells (Fig. S3c). At 7 days after HCT, the percentages of IL-17A$^+$IL-22$^-$ and IL-17A$^-$IL-22$^+$ CD8$^+$ T cells in MLN were higher in recipients given IFN-$\gamma^{-/-}$-SPL cells than in those given WT-spleen cells; and percentages of IL-17A$^+$IL-22$^+$ cells were very low in both groups (Fig. 4c).

**Fig. 3 IL-22 from Th/Tc22 cells causes dysbiosis and bacteria translocation.** Lethally irradiated WT BALB/c recipients were engrafted with WT-TCD-BM alone or combined with splenic T cells from WT or IL-22$^{-/-}$ C57BL/6 donors, and the recipients of splenic T cells were given 1-DEX or 4-DEX treatment as described in Fig. 1. Microbiome profile in feces from the ileum of recipients was measured on day 25 after HCT. **a** Diversity of ileal flora was determined by the number of species, Shannon index, and Fisher Index. Means ± SEM are shown. **b** Principal coordinate analysis of the ileal flora. **c** Bacterial composition at the species level of the ileal flora is depicted with individual mice displayed in each bar. **d** %Abundance of *E. coli*, Enterococcus_ sp._FDAARGOS_553 and Lactobacillus murinus in the ileal flora. Means ± SEM are shown. **e** Bacteria colony numbers in the liver cell suspension culture plates, with one plate per recipient. Means ± SEM are shown. All data combined from two replicate experiments, each dot represents one mouse, $n = 9$ (TCD-BM), $n = 6$ (WT T-1-DEX & WT T-4-DEX & IL-22$^{-/-}$ T-4-DEX). Ordinary one-way ANOVA with Tukey's correction for multiple comparisons was used in **a**, **e**. Kruskal–Wallis test with Dunn's correction for multiple comparisons were performed with two-tailed p value in **d**. **a** Observed: **$p = 0.0022$ (WT-1-DEX vs. WT-4-DEX), **$p = 0.0078$ (WT-4-DEX vs. IL-22$^{-/-}$T-4-DEX); Shannon Index: *$p = 0.0351$; Fisher Index: **$p = 0.0025$ (WT-1-DEX vs. WT-4-DEX), **$p = 0.0077$ (WT-4-DEX vs. IL-22$^{-/-}$T-4-DEX). **d**, **$p = 0.0016$ (1-DEX vs 4-DEX), **$p = 0.0042$ (WT-4-DEX vs. IL-22$^{-/-}$T-4-DEX); **e** ****$p < 0.0001$. T-cell-depleted bone marrow (TCD-BM), dexamethasone (DEX).

In further experiments, sorted CD8$^+$ T cells ($1.5 \times 10^6$) from WT, IFN-$\gamma^{-/-}$ or IFN-$\gamma^{-/-}$ROR$\gamma$t$^{-/-}$ were co-transplanted with WT-TCD-BM cells ($2.5 \times 10^6$). WT CD8$^+$ T cells induced little GVHD, IFN-$\gamma^{-/-}$, and IFN-$\gamma^{-/-}$ROR$\gamma$t$^{-/-}$ T cells both induced severe lethal Gut-aGVHD with diarrhea (Fig. 4d). There was no significant difference in the percentage and yield of IL-17A$^-$IL-22$^+$CD8$^+$ T cells in the MLN of the recipients given IFN-$\gamma^{-/-}$ or IFN-$\gamma^{-/-}$ROR$\gamma$t$^{-/-}$ T cells (Fig. 4e), and they were AHR$^+$ ROR$\gamma$t$^{-/-}$ Tc22 cells (Fig. 4f). These results indicate that Gut-aGVHD-induced by IFN-$\gamma^{-/-}$ donor CD8$^+$ T cells are associated with the expansion of IL-17A$^+$IL-22$^-$Tc17 and AHR$^+$IL-17A$^-$ IL-22$^+$ Tc22 cells.

**IL-22 from Tc22 but not IL-17A from Tc17 is required for Gut-aGVHD induced by IFN-$\gamma^{-/-}$CD8$^+$ T cells.** As Gut-aGVHD induced by IFN-$\gamma^{-/-}$ CD8$^+$ T cells was associated with expansion of IL-17A$^+$IL-22$^-$ Tc17 and IL-17A$^-$IL-22$^+$ Tc22 cells (Fig. 4c), we tested whether IL-17A is required for Gut-aGVHD pathogenesis, by comparing CD8$^+$ T cells from IFN-$\gamma^{-/-}$ and IFN-$\gamma^{-/-}$IL-17A$^{-/-}$ donors. Recipients were engrafted with CD8$^+$ T cells and TCD-BM cells from IFN-$\gamma^{-/-}$ or IFN-$\gamma^{-/-}$IL-17$^{-/-}$ C57BL/6 donors. The non-GVHD recipients given IFN-$\gamma^{-/-}$ or IFN-$\gamma^{-/-}$/IL-17A$^{-/-}$ TCD-BM cells alone were combined into a TCD-BM group (Fig. 5a). Recipients given IFN-$\gamma^{-/-}$ or IFN-$\gamma$/IL-17A$^{-/-}$ CD8$^+$ T cells both developed diarrhea, bodyweight loss, and most of them died within 30 days after HCT, with no difference between the two groups (Fig. 5a). The percentages of IL-17A$^-$IL-22$^+$CD8$^+$ T cells in the MLN at 7 days after HCT did not differ between the two groups (Fig. 5b). These results indicate that IL-17A produced by Tc17 cells is not required for Gut-aGVHD induced by IFN-$\gamma^{-/-}$CD8$^+$ T cells.

To evaluate the role of IL-22, recipients given IFN-$\gamma^{-/-}$CD8$^+$ T cells were treated with anti-IL-22 or control mouse-IgG1 at 150 µg every other day, until day 8 after HCT. Anti-IL-22 treatment effectively prevented diarrhea in all recipients, and the recipients all survived for >30 days (Fig. 5c) with little colon tissue damage (Fig. S4). Anti-IL-22 treatment did not change the percentages of IL-17A$^+$IL-22$^-$ or IL-17A$^-$IL-22$^+$ subsets in the MLN of recipients given IFN-$\gamma^{-/-}$CD8$^+$ T cells (Fig. 5d). Neutrophil infiltration plays an important role in Gut-aGVHD damage[43,44], and IL-22 from T cells attracted neutrophil into tumor tissues[45]. Consistently, neutralizing IL-22 markedly reduced the percentage and numbers of Ly6G$^+$CD11b$^+$ neutrophils in the colon tissue of Gut-aGVHD recipients (Fig. 5e).

In BALB/c recipients treated with neutralizing antibody against IFN-$\gamma$ during the first 5 days after HCT, WT CD8$^+$ T cells induced severe Gut-aGVHD and bacterial translocation but IL-22$^{-/-}$ CD8$^+$ T cells did not (Fig. S5a, b), indicating IL-22 from donor Tc22 cells is required for pathogenesis. Recipient ILC3 in the small intestine and colon were eliminated by 5–7 days after HCT before Gut-aGVHD onset (Fig. S6), suggesting they are

not required for pathogenesis. Therefore, in the absence of IFN-$\gamma$, IL-22 from donor-type Tc22 cells has a critical role in Gut-aGVHD induced by CD8$^+$ T cells.

**Tc22 differentiation from alloreactive IFN-$\gamma^{-/-}$CD8$^+$ T cells requires tissue-damage by conditioning.** Tissue damage from the conditioning regimen before HCT results in production of proinflammatory cytokines such as IL-6 and IL-1$\beta$[46]. As IL-6 plays an important in augmenting Th/Tc22 differentiation[40,47], we tested whether TBI was required for induction of Gut-aGVHD by IFN-$\gamma^{-/-}$CD8$^+$ T cells. To avoid rejection of donor cells in non-conditioned recipients, we used a parent into F1 HCT model using C57BL/6 (H-2$^b$) donors and C57BL/6 x BALB/c (H-2$^{b/d}$) F1 (CB6F1) recipients. Accordingly, CD4$^+$ T-depleted spleen cells containing $2 \times 10^6$ CD8$^+$ T cells from IFN-$\gamma^{-/-}$ donor and TCD-BM ($2.5 \times 10^6$) from WT donor were transplanted into lethal TBI-conditioned or non-conditioned CB6F1 recipients. In addition, the lethal TBI-conditioned recipients were treated with anti-IL-22 mAb or mouse IgG1. IFN-$\gamma^{-/-}$CD8$^+$ T cells induced Gut-aGVHD in the colon but not in the small intestine of lethal TBI-conditioned CB6F1 recipients and the disease was prevented by anti-IL-22 treatment. No evidence of Gut-aGVHD was apparent in non-conditioned recipients (Fig. S7a–c). TBI-conditioning also augmented expansion of the IL-17A$^+$IL-22$^-$ and especially the IL-17A$^-$IL-22$^+$ CD8$^+$ T-cell subsets in the MLN of Gut-aGVHD recipients (Fig. S7d). Taken together, alloreactive Tc22 differentiation from IFN-$\gamma^{-/-}$CD8$^+$ T cells requires tissue damage by conditioning before HCT.

**IL-22 from Tc22 cells induces Gut-aGVHD via host-tissue production of Reg3$\gamma$.** IL-22 can augment Paneth cell and intestinal epithelial cell production of Reg3$\gamma$[4,48]. Consistently, we observed that expression of Reg3$\gamma$ mRNA in ileal tissue and serum Reg3$\gamma$ was higher in recipients given IFN-$\gamma^{-/-}$CD8$^+$ T cells than in those given WT CD8$^+$ T cells on days 7 and 14 after HCT (Fig. 5f, g). Anti-IL-22 treatment markedly reduced ileal tissue expression of Reg3$\gamma$ mRNA and serum concentrations of Reg3$\gamma$ (Fig. 5h). Therefore, we tested whether elevation of Reg3$\gamma$ was required for Gut-aGVHD induced by IFN-$\gamma^{-/-}$ donor CD8$^+$ T cells by transplanting IFN-$\gamma^{-/-}$CD8$^+$ T cells from BALB/c donors into WT or Reg3$\gamma^{-/-}$ C57BL/6 recipients. Tc22 expansion did not differ between WT and Reg3$\gamma^{-/-}$ recipients (Fig. 5i). Expression of Reg3$\gamma$ mRNA in ileal tissue was detected in WT but not Reg3$\gamma^{-/-}$ recipients (Fig. 5j). WT recipients showed Gut-aGVHD, and nearly 90% (7/8) died by 10 days after HCT, but Reg3$\gamma^{-/-}$ recipients showed no signs of Gut-aGVHD, and all survived for >30 days (Fig. 5k). In addition, we treated WT and Reg3$\gamma^{-/-}$ C57BL/6 recipients with 4-DEX after HCT, and we observed significant reduction of Gut-aGVHD in Reg3$\gamma^{-/-}$ recipients (Fig. S8). Therefore, IL-22 from donor Tc22

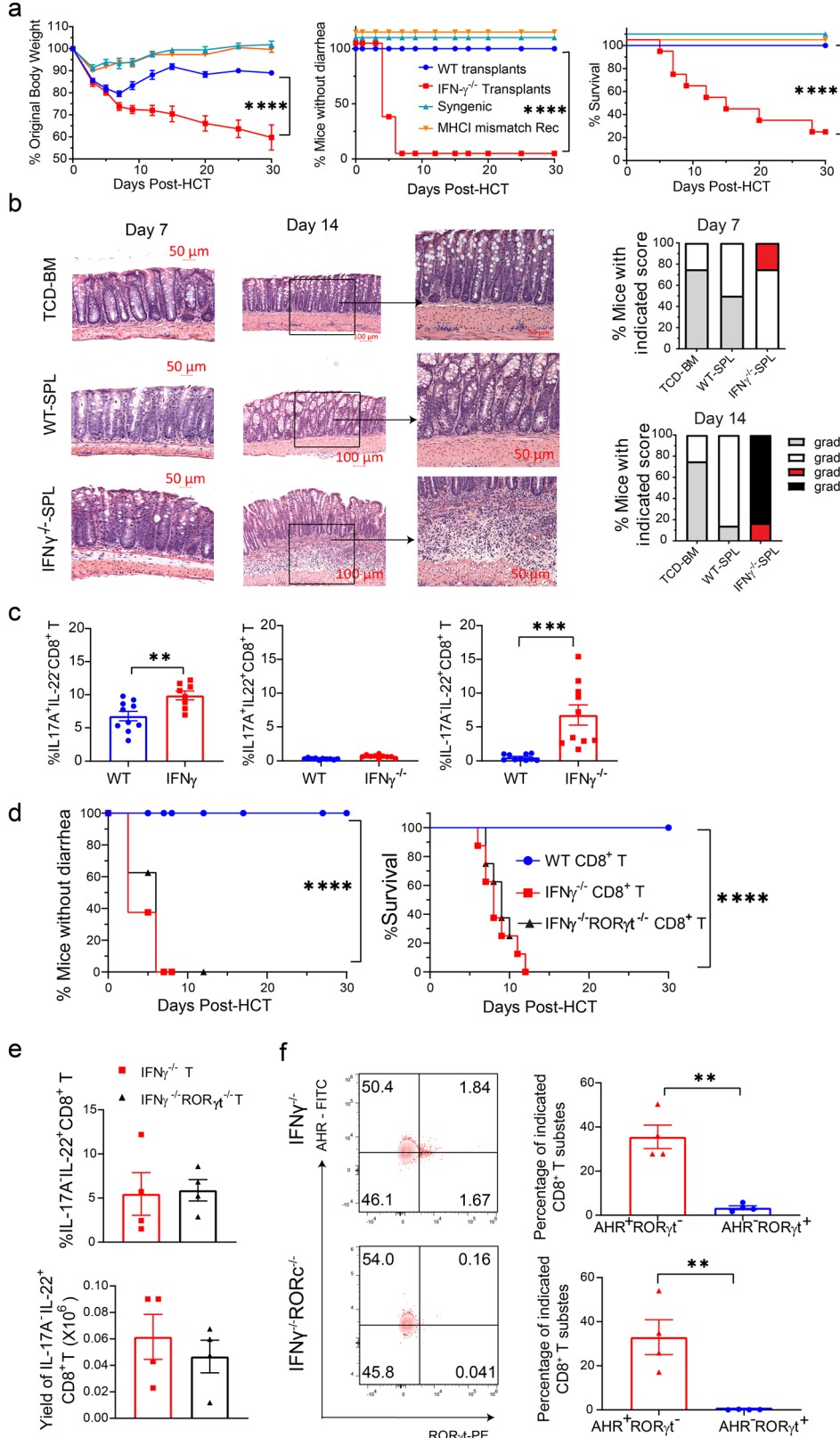

cells augments host-tissue production of Reg3γ, and Tc22 induction of Gut-aGVHD is Reg3γ-dependent.

**IL-22 from Tc22 cells causes dysbiosis via Reg3γ.** IL-22 could cause dysbiosis enhanced by Reg3γ under inflammatory conditions[12]. Therefore, we analyzing 16 S ribosomal RNA sequences[6]

to evaluate the impact of IL-22 and the associated increased production of Reg3γ on microbiota profiles in the ileum of recipients given donor splenic IFN-γ[−/−]-CD8[+] T cells. At 6 days after HCT, the microbiota profile of Gut-aGVHD-free recipients given WT CD8[+] T cells had some differences when compared with untreated BALB/c mice, with increased prevalence of

**Fig. 4 Gut-aGVHD induced by IFN-γ$^{-/-}$ donor CD8$^+$ T cells is associated with expansion of Tc17 and Tc22 cells.** Lethally irradiated WT BALB/c recipients were engrafted with splenocytes containing $1.5 \times 10^6$ T cells combined with TCD-BM ($2.5 \times 10^6$) from WT or IFN-γ$^{-/-}$ C57BL/6 donors. Additional controls include IFN-γ$^{-/-}$ C57BL/6 grafts in syngeneic WT C57BL/6 recipients and MHC I-matched H-2K$^b$MHC-IA$^-$-IE$^-$BALB/c recipients. Allogeneic recipients were also treated with depleting anti-CD4 mAb (500 μg/mouse) immediately after HCT (**a–c**). **a** Mean ± SEM of %Original bodyweight at each time point and recessive curves of %mice without diarrhea and %survival among total mice are shown. $n = 8$ (syngeneic & MHC I-matched recipients), 10 (IFN-γ$^{-/-}$ CD8$^+$ & WT CD8$^+$). **b** Seven and 14 days after HCT, histopathology of colon was evaluated. Representative micrographic photos and %mice with indicated colitis histopathology scores are shown; $n = 4$ biologically independent mice per group. **c** At day 7 after HCT, MLN of recipients were analyzed for donor-type IL-17A$^+$IL-22$^-$, IL-17A$^+$IL-22$^+$ or IL-17A$^-$IL-22$^+$ CD8-T-cell subsets. Means ± SE of percentages are shown, $n = 10$. **d–f** Lethally irradiated WT BALB/c recipients were engrafted with sorted CD8$^+$ T ($1.5 \times 10^6$) from IFNγ$^{-/-}$ or IFNγ$^{-/-}$/RORγt$^{-/-}$ C57BL/6 donors combined with TCD-BM ($2.5 \times 10^6$) from WT C57BL/6 donors. **d** Recessive curves of %mice without diarrhea and %survival among total mice are shown. $n = 8$ from two replicate experiments. **e, f** On day 7 after HCT, MLN cells were analyzed for %IL17A$^-$IL-22$^+$ CD8$^+$ T cells (**e**) and %AHR$^+$ RORγt$^-$ or AHR$^-$RORγt$^+$ among IL17A$^-$IL-22$^+$CD8$^+$ T cells (**f**) means ± SEM are shown, $n = 4$. All results combined from two replicates. Each dot represents one mouse. Log-rank test was performed with two-tailed $p$ value for survival comparisons. Nonlinear regression (curve fit) was used and a two-tailed $p$ value was calculated for bodyweight and diarrhea comparisons. Unpaired two-tailed Student' $t$ test was used to compare means. **a** ****$p < 0.0001$; **c** **$p = 0.0063$, ***$p = 0.0006$; **d** ****$p < 0.0001$. **e** **$p = 0.0010$ (IFNγ$^{-/-}$), **$p = 0.0059$ (IFNγ$^{-/-}$/RORγt$^{-/-}$). T-cell-depleted bone marrow (TCD-BM), SPL (spleen), MLN (mesenteric lymph node).

*Streptococcus* and *E. coli* (Fig. 6a and Fig. S9), although they did not show clinical signs of Gut-aGVHD (Fig. 4a). In contrast, the profiles of Gut-aGVHD recipients given IFN-γ$^{-/-}$CD8$^+$ T cells showed marked changes, with a lower prevalence of protective Clostridiaceae and a higher prevalence of pathogenic *Streptococcus* and *E. coli* (Fig. 6a and Fig. S9). Prevention of Gut-aGVHD with anti-IL-22 treatment was associated with a higher prevalence of protective Clostridiaceae and Lactobacillus as well as a lower prevalence of pathogenic *Streptococcus*, and *E. coli*. (Fig. 6a and Fig. S9). In addition, cultures of liver tissue suspension from GVHD-free recipients of WT CD8$^+$ T cells showed very little bacterial growth, but 37% (17/45) cultures from recipients of IFN-γ$^{-/-}$CD8$^+$ T cells showed exuberant bacterial growth, and anti-IL-22-treatment markedly reduced the frequencies of bacterial growth (Fig. 6b). Bacteria in the liver tissue cultures from recipients given IFN-γ$^{-/-}$CD8$^+$ T cells were predominantly *E. coli* with some *Lactobacillus* and other strains (Fig. 6c).

Finally, the prevalence of protective *Barnesiella* and *Blautia* was higher in Gut-aGVHD-free Reg3γ$^{-/-}$ recipients than in Gut-aGVHD WT recipients, while the prevalence of pathogenic *E. coli* was lower (Fig. 6d). Bacterial growth frequency was also lower in the liver tissue cultures from Reg3γ$^{-/-}$ recipients (Fig. 6e). These results indicate that IL-22 from Tc22 cells causes dysbiosis by augmenting Reg3γ production.

**Dysbiosis is required for induction of Gut-aGVHD-mediated by Tc22-derived from IFN-γ$^{-/-}$CD8$^+$ T cells.** To validate the impact of dysbiosis on induction of Gut-aGVHD, we housed recipients of IFN-γ$^{-/-}$CD8$^+$ T cells separately or together with recipients of WT CD8$^+$ T cells at a ratio of 2:3 in cages of five mice. Recipients of IFN-γ$^{-/-}$CD8$^+$ T cells housed separately all developed diarrhea and bodyweight loss, and 83% (10/12) died by 30 days after HCT (Fig. 7a). In contrast, recipients of IFN-γ$^{-/-}$CD8$^+$ T co-housed with non-GVHD recipients of WT CD8$^+$ T cells showed significant reduction in Gut-aGVHD severity with reduction of diarrhea and higher bodyweight and survival (Fig. 7a) and lower prevalence of *E. coli* but higher prevalence of *Lactobacillus* and *Clostridiaceae* (Fig. 7b). These results suggest that the microbiome from non-GVHD recipients can ameliorate Gut-aGVHD in recipients with dysbiosis.

We further evaluated the role of dysbiosis in Gut-aGVHD pathogenesis by eliminating intestinal bacteria. Recipients of IFN-γ$^{-/-}$CD8$^+$ T cells were given a daily gavage of four antibiotics, including ampicillin (1 g/L), neomycin (1 g/L), metronidazole (1 g/L), and vancomycin (0.5 g/L) (4ABX)[49]. Consistent with a previous report[50], bacteria could not be detected in the feces from

ileum on day 6 by V4-V5 16S amplification. Although phosphate-buffered saline (PBS) control mice all developed diarrhea, and 80% died within 30 days after HCT, the 4ABX treatment completely prevented diarrhea, and all recipients survived for >30 days (Fig. 7c). These results indicate that dysbiosis is required for induction of Gut-aGVHD mediated by IFN-γ$^{-/-}$CD8$^+$ T cells.

**Depletion of donor-type CX3CR1$^{hi}$ MNP is associated with Gut-aGVHD mediated by IFN-γ$^{-/-}$ donor CD8$^+$ T cells.** We found bacterial translocation into the liver in Gut-aGVHD recipients of IFN-γ$^{-/-}$CD8$^+$ T cells but not in non-GVHD recipients of WT CD8$^+$ T cells, although both showed an increased prevalence of *E. coli* (Fig. 6a). In addition, recipients of WT CD8$^+$ T cells did not show signs of Gut-aGVHD when co-housed with recipients of IFN-γ$^{-/-}$CD8$^+$ T cells (Fig. 7a), suggesting a protective mechanism in those mice. Therefore, abnormalities other than dysbiosis may also contribute to the induction of Gut-aGVHD in recipients of IFN-γ$^{-/-}$CD8$^+$ T cells.

CX3CR1$^{hi}$ MNP cells in the gut tissue have stronger ability to downregulate inflammatory responses and prevent bacterial translocation as compared to CX3CR1$^{lo}$ MNP cells[19,51]. Therefore, we explored whether there were any abnormal changes in CX3CR1$^{hi}$ MNP during IFN-γ$^{-/-}$CD8$^+$ T cell-mediated Gut-aGVHD pathogenesis. By 10 days after HCT, CX3CR1$^+$ cells were all donor-type, and the CX3CR1$^+$ cells included CX3CR1$^{lo}$ and CX3CR1$^{hi}$ populations (Fig. S10). The CX3CR1$^{hi}$ MNP in the colon tissues of recipients of WT CD8$^+$ or IFN-γ$^{-/-}$CD8$^+$ T cells expressed higher levels of CD11c, CD11b, F4/80, CD64, MerTK, IL10R, and CSF1-R, as compared with CX3CR1$^{lo}$ MNP (Fig. S10). The percentages and yields of CX3CR1$^{hi}$ MNP in the colon tissue were markedly lower in recipients given IFN-γ$^{-/-}$CD8$^+$ T cells than in those given WT CD8$^+$ T cells (Fig. 8a). The CX3CR1$^{hi}$ MNP in recipients of IFN-γ$^{-/-}$CD8$^+$ T cells expressed higher levels of PD-1 (Fig. 8b) and showed higher percentages of apoptotic Annexin V$^+$ cells (Fig. 8c). Blockade of PD-1 interaction with PD-L1 by anti-PD-L1 mAb significantly increased the percentage of CX3CR1$^{hi}$ MNP in recipients of IFN-γ$^{-/-}$CD8$^+$ T cells (Fig. 8d), although intestinal epithelial cells of those recipients expressed lower levels of PD-L1, as compared with recipients of CD8$^+$ T cells (Fig. S11a, b). Anti-IL-22 had no effect on the yield of CX3CR1$^{hi}$ MNP or their expression of PD-1 in recipients given IFN-γ$^{-/-}$CD8$^+$ T cells (Fig. 8e). These results indicate that Gut-aGVHD induced by IFN-γ$^{-/-}$CD8$^+$ T cells is associated with lower numbers of CX3CR1$^{hi}$ MNP in the colon tissue independent of Tc22 expansion.

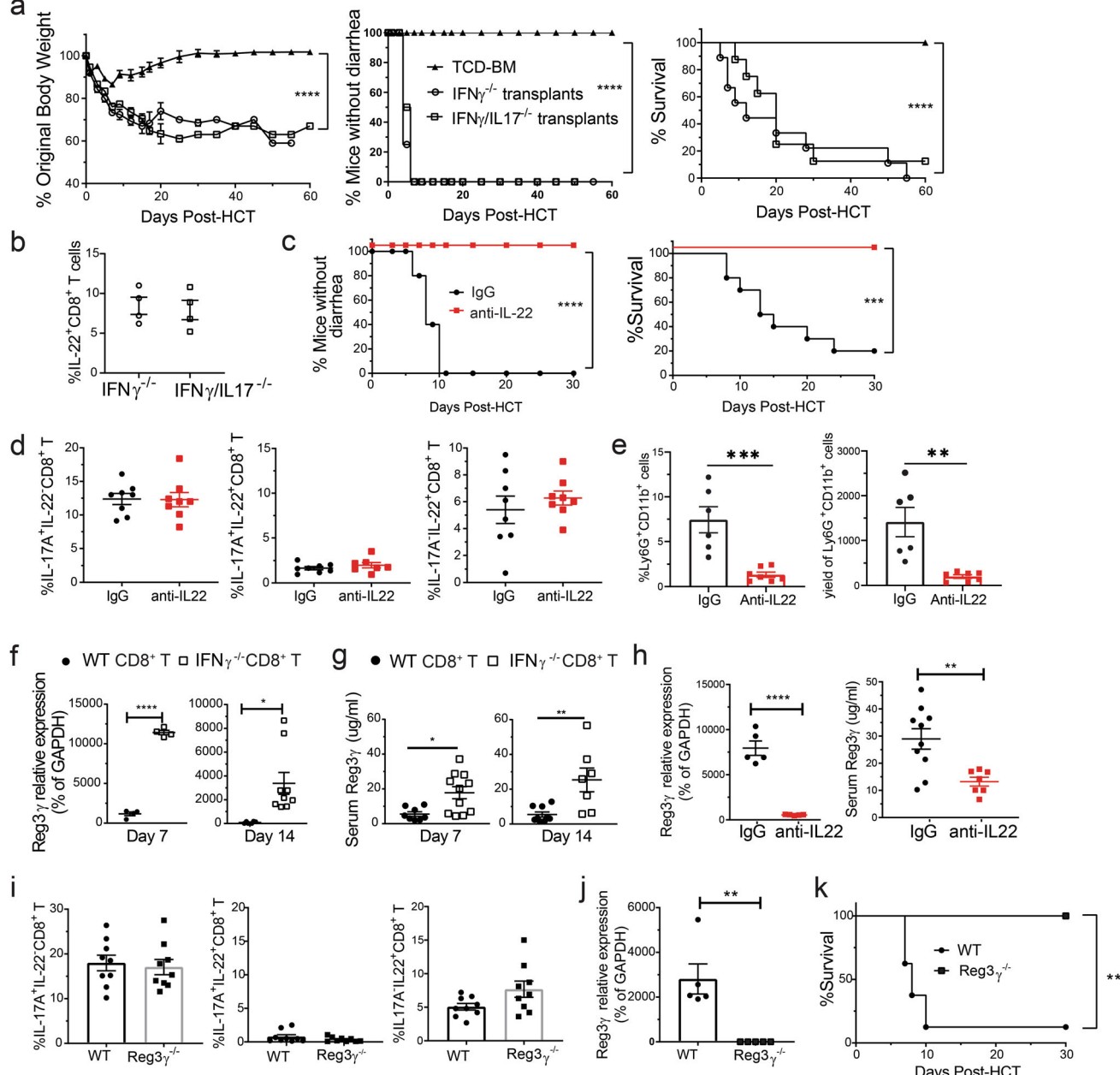

**Fig. 5 Gut-aGVHD induced by IFN-γ⁻/⁻ donor CD8⁺ T cells is Tc22-dependent. a, b** Gut-aGVHD was induced in BALB/c recipients with grafts from IFN-γ⁻/⁻ or IFN-γ⁻/⁻/IL-17⁻/⁻ C57BL/6 donors as described in Fig. 4a. **a** Plots of %mice without diarrhea, %Original bodyweight and %survival. n = 8. **b** Means ± SEM of percentages of IL-17A⁻IL-22⁺ CD8⁺ T cells in MLN at 7 days after HCT are shown. n = 4. **c-e** WT BALB/c recipients with gut-aGVHD induced with grafts from IFNγ⁻/⁻ C57BL/6 donors were treated with anti-IL-22 mAb or control IgG. **c** Plots of %mice without diarrhea and %survival; n = 10. **d** Mean ± SEM of %IL-17A⁺IL-22⁻, IL-17A⁺IL-22⁺, or IL-17A⁻IL-22⁺ CD8⁺ T-cell subsets in MLN at day 7. n = 8. **e** % and yield of neutrophils in the colon at day 10. n = 6 (IgG), 7 (anti-IL-22). **f, g** WT BALB/c with grafts from WT or IFNγ⁻/⁻ C57BL/6 donors were measured for ileal expression of Reg3γ and serum Reg3γ at days 7 and 14. Mean ± SEM, n = 4 (**f**, d7), 6 (**f**, d14, WT), 9 (**f**, d14, IFNγ⁻/⁻ & **g**, d14, WT), 8 (**g**, d7, WT), 11 (**g**, d7, IFNγ⁻/⁻), 7 (**g**, d14, IFNγ⁻/⁻). **h** Mean ± SEM of Ileal Reg3γ mRNA and serum Reg3γ in anti-IL-22 and IgG groups at day 7. n = 5 (mRNA), 10 (Serum, IgG), 7 (Serum, anti-IL-22). **i-k** Lethally irradiated WT or Reg3γ⁻/⁻ C57BL/6 recipients were engrafted with WT-TCD-BM (2.5 × 10⁶) and CD8⁺ T cells (2.5 × 10⁶) from IFN-γ⁻/⁻ BALB/c donors, and recipients for monitored for Gut GVHD and survival. **i** Day 7, Mean ± SEM of %IL-17A⁺IL-22⁻, IL-17A⁺IL-22⁺ or IL-17A⁻IL-22⁺ CD8⁺ T cells in MLN, n = 9. **j** Day 7, means ± SEM of ileal Reg3γ mRNA expression are shown, n = 5. **k** Plots of %survival, n = 8 (WT), 6 (Reg3γ⁻/⁻). All results are combined from two replicates. Each dot represents one mouse. Nonlinear regression (curve fit) was performed with two-tailed p value for bodyweight and diarrhea comparisons. Log-rank test was used with two-tailed p value for survival comparison. Unpaired two-tailed Student' t test was used to compare means between two groups. **a** ****p < 0.0001; **c** ***p = 0.003, ****p < 0.0001; **e** **p = 0.0022, ***p = 0.0010; **f** ****p < 0.0001, *p = 0.0127; **g** *p = 0.0103, ** p = 0.0063; **h** **p = 0.0047, ****p < 0.0001; **j** **p = 0.0031; **k** **p = 0.0021. T-cell-depleted bone marrow (TCD-BM).

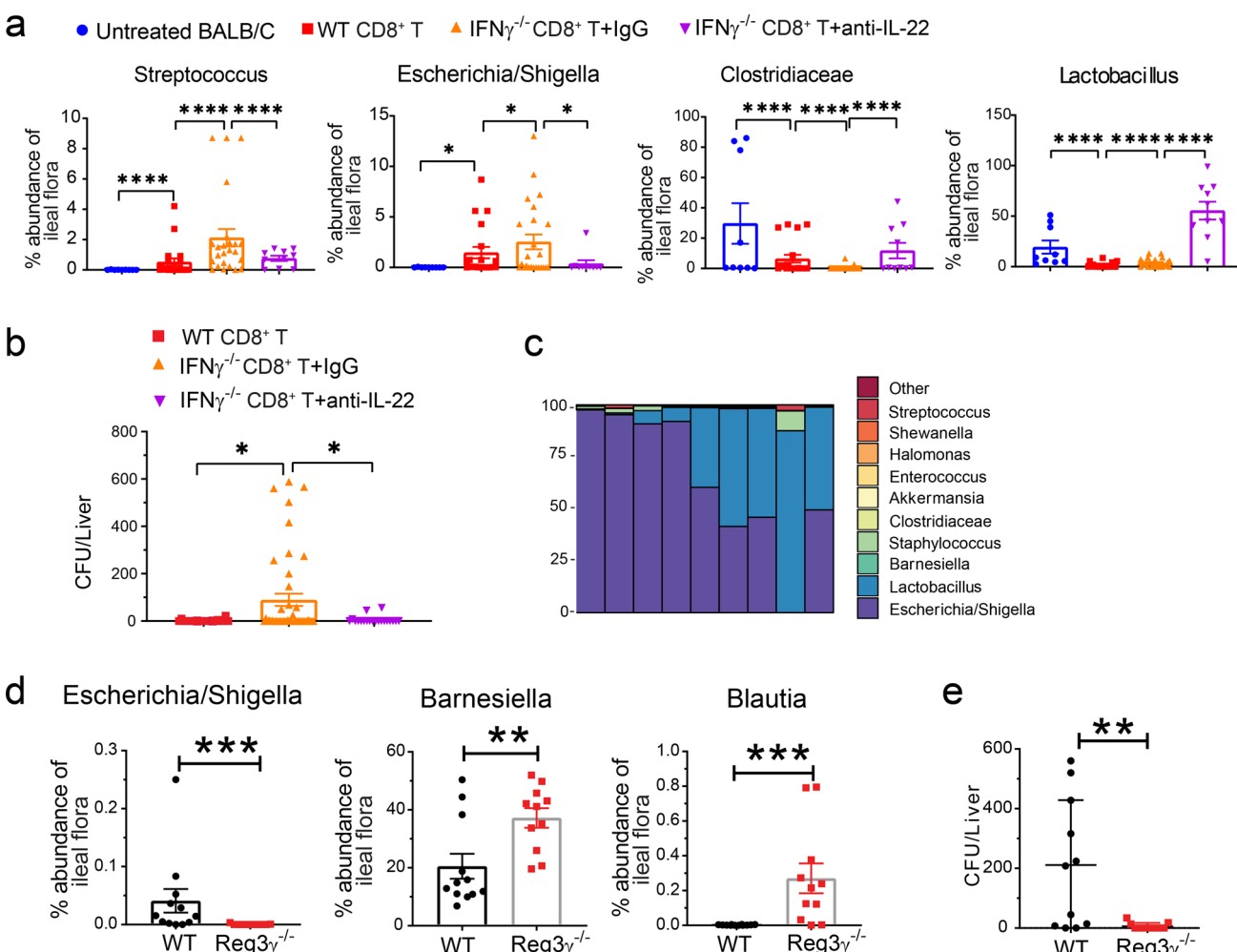

**Fig. 6 Tc22 cells cause dysbiosis via IL-22 and Reg3γ. a–c** Gut-aGVHD was induced in BALB/c recipients with grafts from WT or IFNγ$^{-/-}$ C57BL/6 donors as described in Fig. 4a. Recipients of IFNγ$^{-/-}$ grafts were treated with anti-IL-22 mAb or control IgG (150 μg/mouse) every other day from day 0 to day 6 after HCT. Feces from the ileum of untreated BALB/c, recipients of WT CD8$^+$ T, and recipients of IFNγ$^{-/-}$ CD8$^+$ T treated with anti-IL-22 or control IgG were analyzed for microbiome profiles with 16 S RNA-seq on day 6. **a** %Abundance of *Clostridiaceae*, *Streptococcus*, *Lactobacillus*, and *Escherichia/ Shigella*. Means ± SEM are shown, n = 9 (untreated BALB/c, three replicates), 20 (WT CD8$^+$ T, four replicates), 24 (IFNγ$^{-/-}$ CD8$^+$ T + IgG, five replicates), 10 (IFNγ$^{-/-}$ CD8$^+$ T + anti-IL-22, 2 replicates). **b**, **c** Liver suspensions were cultured on three blood agar plates for 24 hours. Means ± SEM of enumerated bacteria colonies from each liver are shown in **b**. n = 20 (WT CD8$^+$ T & IFNγ$^{-/-}$ CD8$^+$ T + anti-IL-22, four replicates), 45 (IFNγ$^{-/-}$ CD8$^+$ T + IgG, nine replicates). Genus-level bacterial composition of colonies grown from liver suspension of IFNγ$^{-/-}$CD8$^+$ T recipients are displayed in each bar are shown in **c**, n = 9. **d**, **e** Lethally irradiated WT or Reg3γ$^{-/-}$ C57BL/6 recipients were engrafted with IFNγ$^{-/-}$ BALB/c CD8$^+$ T cells (2.5 × 10$^6$) combined with TCD-BM (2.5 × 10$^6$) from WT BALB/c donors. Six days after HCT, ileal feces were analyzed with bacteria16S RNA sequencing, and liver cell suspensions were cultured. **d** %Abundance of *Barnesiella*, *Escherichia/Shigella*, and *Blautia*. Means ± SEM are shown. n = 11 (Reg3γ$^{-/-}$), 12 (WT). Two replicate experiments. **e** Mean ± SE of bacteria colony numbers for each liver. n = 11. Two replicates. Each dot represents one mouse, P value was determined by Kruskal–Wallis with Dunn's multiple comparisons test with two-tailed p value (**a**), ordinary one-way ANOVA (**b**), Mann–Whitney test with two-tailed p value (**d**) and unpaired two-tailed Student's t test (**e**). **a** *p = 0.0362, ****p < 0.0001; **b** *p = 0.0354 (WT CD8$^+$ T vs. IFNγ$^{-/-}$ CD8$^+$ T + IgG), *p = 0.0443 (IFNγ$^{-/-}$ CD8$^+$ T + IgG vs· IFNγ$^{-/-}$ CD8$^+$ T + anti-IL-22); **d** ***p = 0.0003 (*Escherichia/Shigella*); **p = 0.0070; *** p = 0.0004 (*Blautia*); **e** **p = 0.0053.

**Preservation of donor-type CX3CR1$^{hi}$ MNP reverses Gut-aGVHD induced by IFN-γ$^{-/-}$CD8$^+$ T cells.** We tested whether CX3CR1$^{hi}$ MNP could be preserved in PD-L1$^{-/-}$ recipients and whether such preservation could prevent Gut-aGVHD. At ~7 days after HCT, diarrhea developed in both WT and PD-L1$^{-/-}$ recipients of IFN-γ$^{-/-}$CD8$^+$ T cells, but not in WT recipients of WT CD8$^+$ T cells. WT recipients of IFN-γ$^{-/-}$CD8$^+$ T cells continued to have diarrhea, and most of them died by 30 days after HCT, but PD-L1$^{-/-}$ recipients of IFN-γ$^{-/-}$CD8$^+$ T cells gradually recovered and became diarrhea-free, and all survived for more than 30 days after HCT (Fig. 9a). Recovery in PD-L1$^{-/-}$ recipients was associated with higher percentages and yields of

CX3CR1$^{hi}$ MNP in colon tissue (Fig. 9b) and lower levels bacterial translocation into the liver (Fig. 9c).

To validate the protective role of CX3CR1$^{hi}$ MNP in the Gut-aGVHD recipients, we transplanted TCD-BM cells from WT or CX3CR1$^{-/-}$ donors that cannot generate CX3CR1$^{hi}$ MNP cells together with IFN-γ$^{-/-}$CD8$^+$ T cells into PD-L1$^{-/-}$ recipients. Recipients given WT-donor TCD-BM cells developed diarrhea ~5 days after HCT but spontaneously recovered by 7 days after HCT, and all survived for >15 days after HCT. In contrast, recipients given CX3CR1$^{-/-}$ donor BM cells developed diarrhea without subsequent recovery, and 67% (6/9) of the recipients died within 10 days after HCT (Fig. 9d). The CX3CR1$^{-/-}$-BM

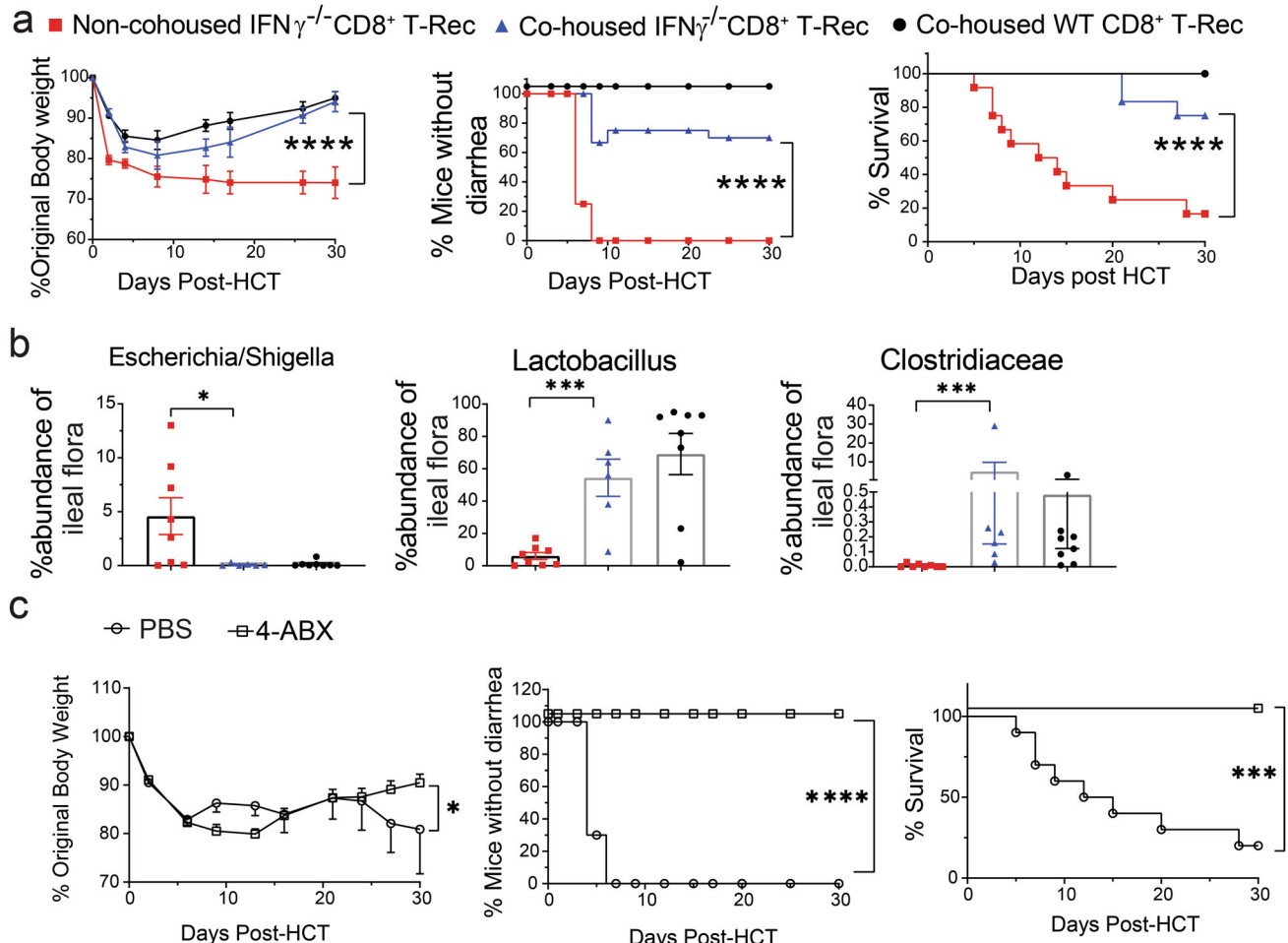

**Fig. 7 Dysbiosis is required for Gut-aGVHD induced by IFNγ$^{-/-}$CD8$^+$ T-derived Tc22 cells.** Gut-aGVHD was induced in BALB/c recipients with grafts from WT or IFNγ$^{-/-}$ C57BL/6 donors as described in Fig. 4a. **a, b** Starting on the day of HCT, recipients of IFNγ$^{-/-}$CD8$^+$T cells were separately housed or co-housed with recipients of WT CD8$^+$ T cells. **a** Mean ± SEM of %Original bodyweight at each time point and recessive curves of %mice without diarrhea and %survival among total mice are shown. n = 12 combined from two replicate experiments. **b** %Abundance of *Escherichia/Shigella*, *Clostridiaceae,* and *Lactobacillus* from ileal fecal samples. Means ± SEM, n = 6 (co-house IFNγ$^{-/-}$ CD8$^+$ T), 8 (Non-co-house IFNγ$^{-/-}$ CD8$^+$ T & WT CD8$^+$ T) from two replicate experiments. **c** Recipients of IFNγ$^{-/-}$CD8$^+$ T cells were gavaged with a mixture of Ampicillin(1 g/L), Neomycin(1 g/L), Metronidazole(1 g/L), and Vancomycin (0.5 g/L) (4ABX) or PBS (250 µl/mouse/day) from days 0 to 7 following HCT.) Mean ± SEM of %original bodyweight at each time point and recessive curves of %mice without diarrhea and %survival among total mice are shown. n = 10 from two replicate experiments. Each dot represents one mouse, *P* value was determined by nonlinear regression (curve fit) (**a**, **c** bodyweight and diarrhea) with two-tailed *p* value, Log-rank test (**a**, **c** survival) with two-tailed *p* value, Kruskal–Wallis test with Dunn's correction for multiple comparisons with two-tailed *p* value **b**. **a** ****p < 0.0001; **b** *p = 0.0463, ***p = 0.0002; **c** *p = 0.0480, ***p = 0.0002.

cells did not restore CX3CR1$^{hi}$ MNP (Fig. 9e). The lack of CX3CR1$^{hi}$ MNP led to greater expansion of Tc22 cells and higher bacterial translocation into the MLN and liver tissues after HCT (Fig. 9f, g).

In further experiments, we transferred CX3CR1$^{hi}$ or CX3CR1$^{lo}$ MNP ($0.5 \times 10^6$) from PD-L1$^{-/-}$ recipients of IFN-γ$^{-/-}$CD8$^+$ T cells at day 10 after HCT into WT recipients of IFN-γ$^{-/-}$CD8$^+$ T cells at day 1 after HCT. Recipients given CX3CR1$^{lo}$ MNP developed diarrhea and lost bodyweight, and most died within 30 days after HCT. Recipients given CX3CR1$^{hi}$ MNP did not develop diarrhea and had less weight loss, all survived for >30 days (Fig. 9h). Therefore, depletion of protective donor-type CX3CR1$^{hi}$ MNP cells is also required for induction of Gut-aGVHD by IFN-γ$^{-/-}$ T cells.

**Depletion of CX3CR1$^{hi}$ MNP contributes to bacterial translocation in SR-Gut-aGVHD.** As DEX treatment reduced

IFN-γ-producing Th1/Tc1 (Fig. 1), and 4-DEX treatment significantly increased bacterial translocation as compared with 1-DEX-treament (Fig. 3e), we tested the impact of 4-DEX treatment on donor-type CX3CR1$^{hi}$ MNP in the colon. At 25 days after HCT, the percentages and yields of donor-type CX3CR1$^{hi}$ MNP cells in the colon tissues were lower in recipients given 4-DEX treatment than in those given 1-DEX treatment (Fig. 10a). Comparing recipients after 1-DEX treatment, Gut-aGVHD was more severe in recipients given CX3CR1$^{-/-}$ BM than in recipients given CX3CR1$^{+/-}$ BM (Fig. 10b). The exacerbation of Gut-aGVHD in recipients given CX3CR1$^{-/-}$ BM was associated with expansion of IL-17A$^-$IL-22$^+$ CD4$^+$ and CD8$^+$ Th/Tc22 cells on day 7 after HCT (Fig. 10c). These results suggest that depletion of donor-type CX3CR1$^{hi}$ MNP cells contributes to SR-Gut-aGVHD pathogenesis and that in the absence of CX3CR1$^{hi}$ MNP cells, 1-DEX treatment augments expansion of donor-type Th/Tc22 cells.

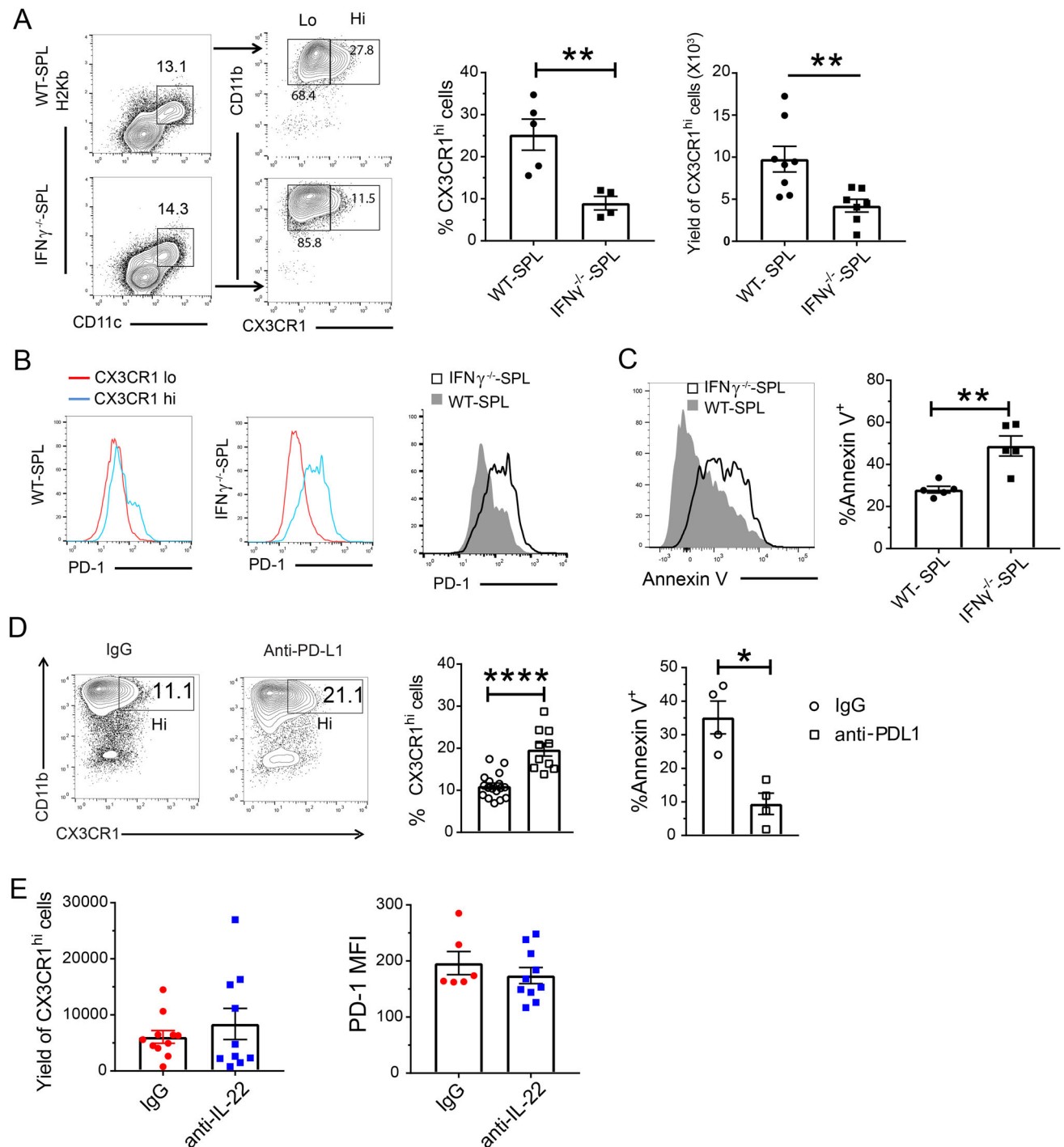

**Fig. 8 Gut-aGVHD induced by IFNγ$^{-/-}$CD8$^+$ T cells is associated with depletion of donor-type CX3CR1$^{hi}$ MNP via PD-1.** Gut-aGVHD was induced in BALB/c recipients with grafts from WT or IFNγ$^{-/-}$ C57BL/6 donors as described in Fig. 4a. **a** Ten days after HCT, colonic lamina propria mononuclear cells were analyzed for percentage and yield of donor-type CD11c$^+$CD11b$^+$CX3CR1$^{hi}$ MNP. Representative patterns and means ± SEM of percentage and yield are shown. $n = 8$. **b** CX3CR1$^{hi}$ and CX3CR1$^{lo}$ MNP from recipients given WT and IFNγ$^{-/-}$ CD8$^+$ T cells are shown in histogram of PD-1 staining; one representative pattern is shown for $n = 5$ in each group. **c** Apoptosis of CX3CR1$^{hi}$ MNP from the recipients given WT or IFNγ$^{-/-}$ CD8$^+$ T cells was measured by Annexin V staining. A representative staining pattern and means ± SEM of percentage of Annexin V$^+$ cells are shown. $n = 5$. **d** Recipients given IFNγ$^{-/-}$ CD8$^+$ T cells were treated with anti-PD-L1 mAb or control IgG (400 μg/mouse) on days 0, 3, and 6 after HCT. On day 10, the percentages of CX3CR1$^{hi}$ MNP and percentages of Annexin V$^+$ CX3CR1$^{hi}$ MNP were measured. One representative staining pattern and means ± SEM are shown. $n = 4$ (% Annexin V$^+$), 10 (%CX3CR1$^{hi}$). **e** Recipients given IFNγ$^{-/-}$ CD8$^+$ T were treated with anti-IL-22 mAb or control IgG (150 μg/mouse) every other day from days 0 to 6 after HCT. On day 10, the yield of CX3CR1$^{hi}$ MNP and their expression of PD-1 were measured. Means ± SEM of percentage of CX3CR1$^{hi}$ MNP and MFI are shown. $n = 10$. All results are combined from two replicate experiments; each dot represents one mouse, unpaired two-tailed Student's $t$ test was used to compare means between two groups. **a** $^{**}p = 0.0039$, $^{****}p < 0.0001$; **c** $^{**}p = 0.0033$; **d** $^{**}p = 0.0045$, $^{***}p = 0.0007$.

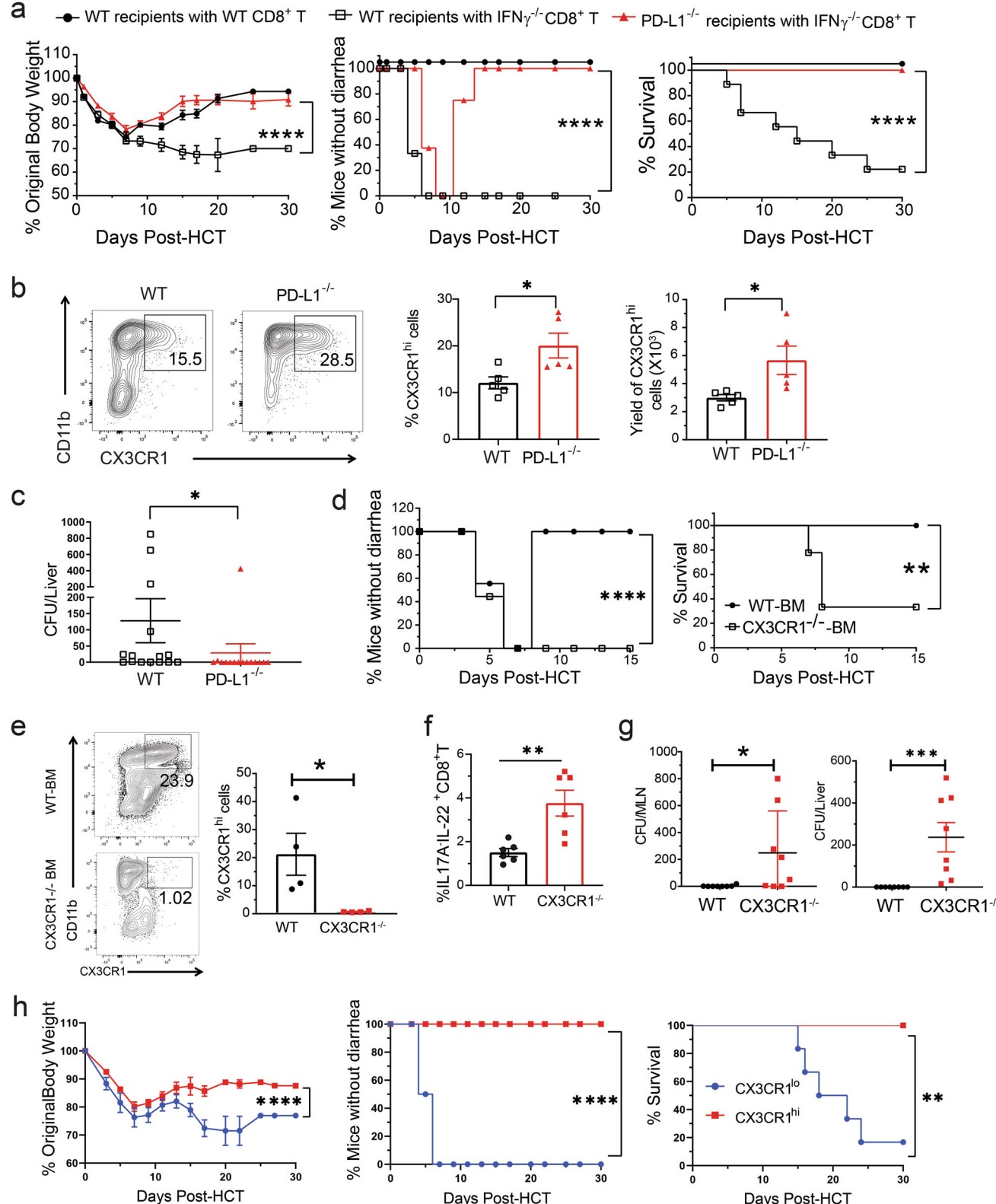

## Discussion

Whether or not donor T cells mediate SR-Gut-aGVHD and how IFN-$\gamma^{-/-}$ donor CD8$^+$ T cells mediate lethal aGVHD are decades-old questions without answers. Using SR-Gut-aGVHD model and murine Gut-aGVHD models induced by IFN-$\gamma^{-/-}$CD8$^+$ T cells, we have implicated two mechanisms: dysbiosis mediated by expansion of Th/Tc22 cells and depletion of CX3CR1$^{hi}$ MNP mediated by PD-1. Our studies provide insights into the pathogenesis of SR-Gut-aGVHD.

First, IFN-$\gamma$ deficiency in donor T cells and prolonged steroid treatment preferentially expand alloreactive Th/Tc22 in the gut tissue of GVHD recipients. We previously reported that alloreactive T cells reciprocally differentiated into Th1, Th2, and Th17, and Th1 and Th17 preferentially infiltrated the gut

**Fig. 9 Gut-aGVHD induced by IFNγ−/−CD8+ T cells is reversed by preservation of donor-type CX3CR1hi MNP. a–c** WT or PD-L1−/− BALB/c recipients were engrafted with IFNγ−/− splenocytes and WT-TCD marrow cells from C57BL/6 donors as described in Fig. 4a. **a** Mean ± SEM of %original bodyweight at each time point and recessive curves of %mice without diarrhea and %survival among total mice are shown. n = 8 (PD-L1−/− recipients with IFNγ−/− CD8+ T), 9 (WT recipients with WT or IFNγ−/− CD8+ T). Two replicate experiments. **b** Percentage and yield of CX3CR1hi MNP in colon tissues on day 10 after HCT. One representative staining pattern and means ± SEM are shown, n = 5. Two replicate experiments. **c** Means ± SEM of bacteria colony numbers in liver suspension cultures from WT or PD-L1−/− recipients 7 days after HCT. n = 15. Three replicate experiments. **d–g** PD-L1−/− BALB/c recipients were transplanted with sorted IFNγ−/−CD8+ T cells with WT or CX3CR1−/− TCD-BM from C57BL/6 donors. Two replicate experiments. **d** Recessive curves of %mice without diarrhea and %survival are shown. n = 9. **e** Representative flow cytometry pattern and %CX3CR1hi MNP in colon tissue. Means ± SEM are shown, n = 4. **f** Mean ± SEM of %IL17A−IL-22+ CD8+ T cells in MLN on day 7, n = 6. **g** Bacteria colony numbers in MLN and liver cell suspension from recipients of WT or CX3CR1−/− TCD-BM cells. Means ± SEM of CFU are shown, n = 8. **h** Splenic CX3CR1hi or CX3CR1lo MNP cells (0.5 × 10^6) from PD-L1−/−BALB/c recipients of IFNγ−/−CD8+ T on day 10 after HCT were sorted and then iv. injected into WT BALB/c recipients of IFNγ−/−CD8+ T cells on day 1 after HCT. The WT Recipients were monitored for clinical signs of aGVHD for up to 30 days, and mean ± SEM of %original bodyweight at each time point and recessive curves of %mice without diarrhea and %survival among total mice are shown. n = 6 from two replicate experiments. Each dot represents one mouse, P value was determined by nonlinear regression (**a**, **d**, **h**, bodyweight & diarrhea) with two-tailed p value, Log-rank test (**a**, **d**, **h**, survival) with two-tailed p value, nonparametric Mann–Whitney test (**c**, **g**) with two-tailed p value, unpaired two-tailed Student' t test (**b**, **e**, **f**). **a** ****p < 0.0001; **b** *p = 0.0271 (%), *p = 0.0329 (Yield); **c** *p = 0.0104; **d** **p = 0.0038, ****p < 0.0001; **e** *p = 0.0336; **f** **p = 0.0044; **g** *p = 0.0286, ***p = 0.0002; **h** **p = 0.0043, ****p < 0.0001.

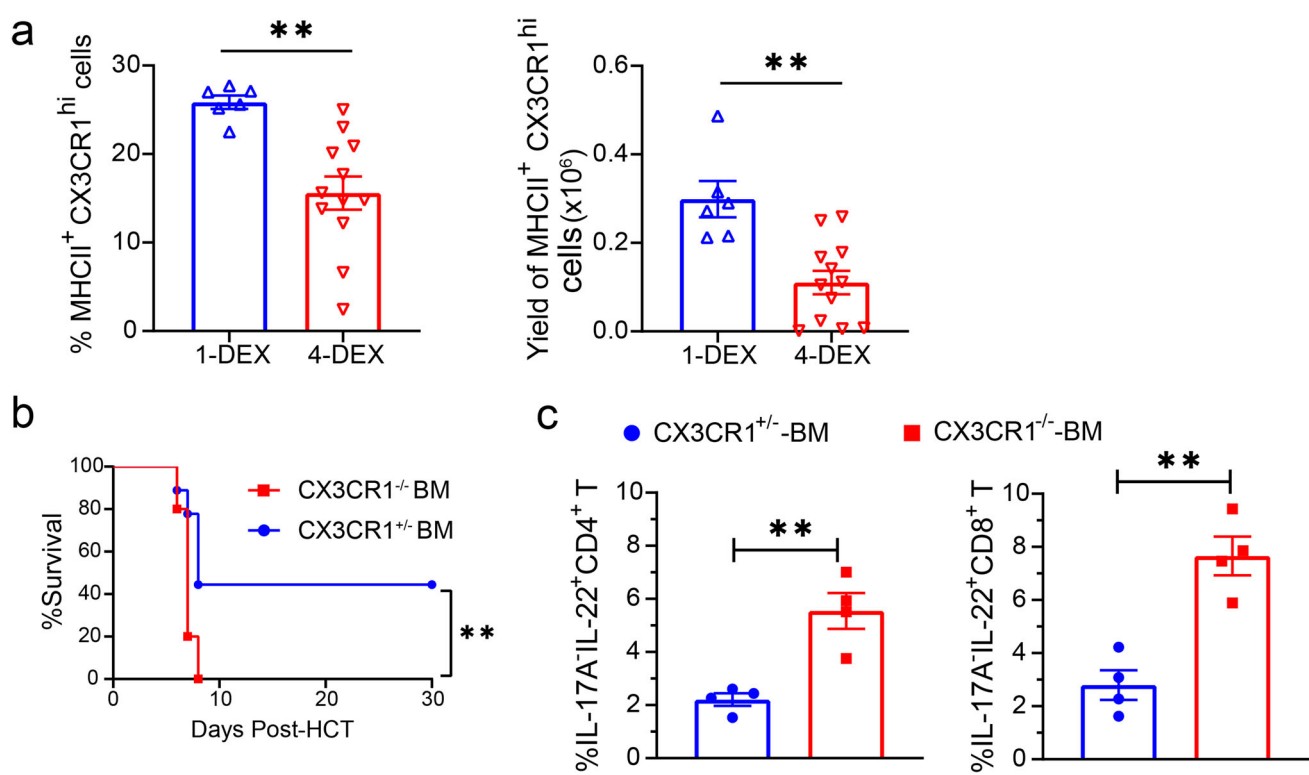

**Fig. 10 SR-Gut GVHD is associated with depletion of CX3CR1hi MNP. a** WT BALB/c recipients were engrafted with spleen and BM cells from WT C57BL/6 donors and were treated with 1-DEX or 4-DEX, as described in Fig.1. Means ± SEM of the percentage and yield of donor-derived (H-2Kb+) CX3CR1hi MNP cells in the colon of DEX-1 or DEX-4 recipients at day 25 after HCT are shown. n = 6 (1-DEX), 12 (4-DEX). **b**, **c** Lethally irradiated WT BALB/c recipients were given TCD-BM (2.5 × 10^6) from CX3CR1+/− or CX3CR1−/− C57BL/6 donors together with splenocytes (5 × 10^6) from CX3CR1+/− C57BL/6 donors. Recipients were given a single i.v. injection of dexamethasone (5 mg/kg) on day 3. **b** %survival, n = 9 (CX3CR1+/− BM), 10 (CX3CR1−/− BM). **c** At day 7 after HCT. Means ± SEM of %IL-17A−IL-22+ cells among CD4+ and CD8+ T cell from MLN are shown. n = 4. All results combined from two replicated experiments. Each dot represents one mouse. Unpaired two-tailed Student' t test was used to compare means between groups. Log-rank test with two-tailed p value was used for survival comparisons. **a** **p = 0.018 (%), **p = 0.011 (yield); **b** **p = 0.0093; **c** **p = 0.0035 (CD4+ T), **p = 0.0018 (CD8+ T). Dexamethasone (DEX).

tissues[52,53]. Others reported that IL-17A+IL-22+ Tc17 cells were a transient lineage occurred only early after HCT[54], although AHR+Tc22 cells have not yet been reported. Consistently, we observed that donor IFN-γ−/−CD8+ T cells preferentially differentiated into RORγt+IL-17A+IL-22− Tc17 and AHR+IL-17A−IL-22+ Tc22 cells in the TBI-conditioned but not non-conditioned recipients. The expansion of Tc22 may result from increased release of IL-6 and IL-1β from TBI-damaged tissues[46]

that augmented Th/Tc22 differentiation and from reduction of T-bet expression in IFN-γ−/− T cells that inhibit Th/Tc22 differentiation[17]. Similarly, we observed that prolonged steroid treatment also preferentially augmented Th/Tc22 expansion but reduced Th/Tc17 expansion. This may be owing to that steroid treatment reduced tissue release of TGF-β but not IL-6[37]. Reduction of TGF-β could inhibit Th/Tc17 differentiation or expansion but augment Th/Tc22 expansion[17,40]. Reduction of

IFN-$\gamma^+$ Th1/Tc1 could also augment expansion of Th/Tc22 by reducing T-bet-mediated inhibition[17]. Interestingly, steroid-resistant colitis is associated with high serum IL-6 concentrations[37]. Thus, it is of interest to test whether blockade of IL-6R signaling reduces Th/Tc22 expansion and prevent induction of SR-Gut-aGVHD.

Second, expansion of Th/Tc22 causes dysbiosis and Gut-aGVHD. Host cell-derived IL-22 augmented intestinal epithelial stem cell and Paneth cell survival and expansion[8,32] and control intestinal microbiome homeostasis via augmenting Reg3γ secretion[47,55]. Gut-aGVHD caused by Th1 and Tc1 cells resulted in reduction of Reg3γ, dysbiosis and exacerbation of Gut-aGVHD[6,7,10]. In contrast, over-production of IL-22 and Reg3γ under inflammatory situation was also found to augment pathogen colonization in the gut tissues[12]. Consistently, we observed that Gut-aGVHD caused by expansion of Th/Tc22 cells was associated with little or reduced damage in small intestinal epithelium and Paneth cells in the recipients of IFN-$\gamma^{-/-}$CD8$^+$ T cells or recipients with SR-Gut-aGVHD. Instead, the Gut-aGVHD caused by Th/Tc22 cells was associated with severe infiltration of neutrophils in the colon tissues, and the pathogenesis required IL-22$^-$- and Reg3γ-dependent dysbiosis. Our results highlight the point that either under or overproduction of IL-22 and Reg3γ can induce dysbiosis that contributes to the pathogenesis of Gut-aGVHD. Interestingly, although positive response with administration of IL-22 was 100% in low-risk Gut-aGVHD patients, it was <50% in high-risk patients[56]. Patients with SR-Gut-aGVHD belongs to high-risk[27]. Consistent with pathogenic role of neutrophils[43,44], we observed that anti-IL-22 treatment markedly reduced neutrophil infiltration in the colon tissues of Gut-aGVHD induced by Tc22 cells. Therefore, our studies suggest a need for caution in clinical testing of IL-22 as a therapy for Gut-aGVHD or colitis, especially those patients with potential SR-Gut-aGVHD or colitis. In addition, IL-22 antagonist may ameliorate SR-Gut-aGVHD.

Third, we observed that IFN-γ deficiency in donor T cells as well as reduction of IFN-$\gamma^+$ Th/Tc1 cells by steroid treatment led to depletion of CX3CR1$^{hi}$ MNP in the gut tissue of GVHD recipients; in addition, lack of donor T-derived IFN-γ led to CX3CR1$^{hi}$ MNP upregulating expression of PD-1 and becoming more sensitive to tissue PD-L1-induced apoptosis. Our studies indicate that donor T-derived IFN-γ is a double-edged sword that can either augment Gut-aGVHD by direct damage to Paneth and intestinal stem cells[57] or reduce Gut-aGVHD by augmenting expansion of protective CX3CR1$^{hi}$ MNP. Our studies call for caution in testing combinations of agents that concurrently inhibit IFN-γR and JAK2 signaling[58]. In addition, IFN-γR agonist may ameliorate SR-Gut-aGVHD by reducing apoptosis of CX3CR1$^{hi}$ MNP and restoring their protective function.

Finally, we observed that either elimination of intestinal bacterial or preservation/transfer of CX3CR1$^{hi}$ MNP prevented induction of Gut-aGVHD induced by expansion of Th/Tc22 cells. Therefore, as depicted in the diagram (Fig. S12), we theorize that simultaneous dysbiosis triggered by expansion of Th/Tc22 and reduction of protective CX3CR1$^{hi}$ MNP associated with lack of donor T-derived IFN-γ result in SR-Gut-aGVHD. These studies open an avenue towards understanding pathogenesis and developing approaches for preventing and treating SR-Gut-aGVHD.

## Methods

**Mice.** BALB/c (H-2$^d$) (Stock# 555) and C57BL/6 (H-2$^b$) (Stock# 556) mice were purchased from the National Cancer Institute (Frederick, MD). PD-L1$^{-/-}$ BALB/c (H-2$^d$) breeders were provided by Dr. Lieping Chen (Yale University)[59,60]. Reg3γ$^{-/-}$ C57BL/6 (H-2$^b$) breeders were provided by Dr. James Ferrara (Mount Sinai Hospital, NY)[33]. IFN-γ$^{-/-}$ C57BL/6 (H-2$^b$) (Stock# 002287), IFN-γ$^{-/-}$ BALB/c (H-2$^d$) (Stock# 002286), CB6F1 (H-2$^{b/d}$) (Stock# 100007), IL-22$^{-/-}$ C57BL/6 (H-2$^b$) (Stock# 027524), B6(Cg)-Rorc$^{tm3Litt}$/J (Stock# 008771), B6.Cg-Tg (Cd4-cre)1Cwi/BfluJ (Stock# 022071), Rag2$^-$γc$^-$ (Stock# 014593) and B6.129P2 (Cg)-Cx3cr1$^{tm1Litt}$/J (Stock# 005582) were purchased from the Jackson Laboratory (Bar Harbor, ME). B6.129P2(Cg)-Cx3cr1$^{tm1Litt}$/J mice were mated with WT C57BL/6 mice to generate CX3CR1$^{+/-}$ mice. B6(Cg)-Rorc$^{tm3Litt}$/J mice were mated with B6.Cg-Tg(Cd4-cre)1Cwi/BfluJ mice to generate T- RORγt$^{-/-}$ C57BL/6 (H-2$^b$) mice. RORγt$^{-/-}$ C57BL/6 (H-2$^b$) mice were provided by Dr. Zuoming Sun (City of hope, Duarte)[61]. IFN-γ$^{-/-}$ C57BL/6 mice were crossing with RORγt$^{-/-}$ C57BL/6 mice to generate the IFN-γ$^{-/-}$/RORγt$^{-/-}$ C57BL/6 (H-2$^b$) mice. IFN-γ$^{-/-}$IL-17$^{-/-}$ C57BL/6 (H-2$^b$) mice were generated by crossing IFN-γ$^{-/-}$ mice with IL-17$^{-/-}$ mice[52]. H-2Kb$^+$IA$^-$IE$^-$ BALB/c mice were generated by backcrossing MHCII$^{-/-}$ C57BL/6[62] mice into WT BALB/c mice for >12 generations. Animal breeding and experiments were performed in separate specific pathogen-free rooms, and control and experimental mice were kept in separate cages in the same room at City of Hope Animal Research Center (ARC). In all, 12 light/12 dark cycle, temperatures of 68–75 °F with 30–70% humidity are used. All procedures were performed in the animal facility in compliance with a protocol approved by the City of Hope Institutional Animal Care and Use Committee (IACUC) under IACUC protocol 03008. All Mice were killed by $CO_2$ from compressed gas cylinders, and we complied with all the ethical regulations.

**Murine GVHD model.** In general, male mice were used at 8–12 weeks of age, BALB/c recipients were exposed to 850 cGy total body irradiation in a single fraction, and C57BL/6 and CB6F1 recipients were exposed to 1300 cGy total body irradiation in a single fraction. Splenocytes and T cell depleted BM cells from donors were injected via tail vein into recipients 6–8 hours after irradiation. DEX (5 mg/kg) was given by i.v. injection on day 3 alone or on days 3, 10, 15, and 20 after HCT. Depletion of T cells from the BM was accomplished by using biotin-conjugated anti-CD4 and anti-CD8 mAbs, and streptavidin Microbeads (Miltenyi Biotec, Germany), followed by passage through an autoMACS Pro cell sorter (Miltenyi Biotec, Germany). Microbeads (Ly-2, Miltenyi Biotec, Germany) were used for to purify CD8$^+$ T cell, and purity was >99%. Clinical acute gut GVHD was assessed by diarrhea, bodyweight and survival.

**Establish Xeno-GVHD model.** Rag2$^-$γc$^-$ male mice were used at 8–12 weeks of age. Mice were given a single intravenous injection of clodronate liposomes at 0.1 ml/mouse one day before irradiation and were exposed to 350 cGy total body irradiation in a single fraction before injection of human PBMCs on the same day. Human PBMC were from unidentified healthy donors at City of Hope Blood Transfusion Center with IRB exemption approval under COH protocol#/Ref#: 21011/202418. The Human PBMCs were further processed via Ficoll Paque Plus (GE healthcare) density centrifugation and washed twice in PBS. Cells were then counted and suspended in PBS at $30 \times 10^6$ cells/0.2 ml. Cell suspensions containing $30 \times 10^6$ cells were injected via the tail vein.

**Isolation of lamina propria cells from mouse intestine.** Longitudinal sections the small intestine or colon were processed using a gentleMACS Dissociator and mouse lamina propria dissociation kit (both Miltenyi Biotec), according to the manufacturer's protocol.

**Blockade of mouse interleukin-22 and mouse IFNγ in vivo.** Anti-mouse IL-22 (Genentech Clone 8E11, mouse IgG1) was administered via intraperitoneal (i.p.) injection at 150 μg every 3 days from day 12 to day 21 after HCT (Fig. 2, 3) or every other day from day 0 to day 8 after HCT (Figs. 5, 6, supplemental Figs. 4 and 7). Certain control groups also received isotype control IgG1 mAb. Anti-mouse IFN-γ (Bio-x-cell, clone R4-6A2) was administered via i.p. injection at 1 mg every day from day 0 to day 5 after HCT. Anti-mouse CD4 (Bio-x-cell, clone GK1.5) was administered via i.p. injection at 500 μg on the day of HCT.

**Antibodies, FACS analysis, and FACS sorting.** All the antibody information is provided in supplementary table 1. Annexin V were detected by Annexin V staining kit from ebioscience (catalog# 88-8103-74) according to manufacturer's protocol. Flow cytometry analyses were performed with CyAn Immunocytometry system (DAKO Cytomation, Fort Collins, CO), Attune NxT Flow Cytometer (ThermoFisher Scientific) and BD LSRFortessa (Franklin Lakes, NJ), the resulting data were analyzed with FlowJo software (Tree Star, Ashland, OR). Cell sorting was performed with a BD FACS Aria SORP sorter at the City of Hope FACS facility. The sorted cells were used for transfer experiments.

**RNA isolation and real-time reverse transcriptase PCR.** We isolated tissue RNA with the TRIzol Reagent (Life technology) according to the manufacturer's instruction. We conducted real-time RT-PCR on an ABI 7500 Real-Time PCR system (Applied Biosystems) with primers and power SYBR Green PCR master mix (Applied Biosystems). Samples were normalized to the control housekeeping gene. Real-time RT-PCR analysis of mRNA for Reg3γ, Defensin-α1 and Defensin-α3 was performed as described in previous publications[53]. Primers used are provided in supplementary table 2.

**ELISA**. Reg3γ (antibodies online), ST2 (R&D system), and sTNFR1 (Thermo Scientific) in serum were measured by enzyme-linked immune sorbent assay (ELISA) according to manufacturer's protocol. Serum was collected and spun down at 2000 rpm for 10 min at 4 °C. Clear supernatant was collected and stored at −20 °C until ELISA analysis.

**Sample collection and DNA extraction**. Ileal stool samples were frozen at −80 °C. For 16 S Miseq sequencing, DNA was extracted with the Power Soil DNA isolation kit (QIAGEN).

**16S rRNA gene amplification and Miseq sequencing and data analysis**. For Fig. 6 & 7, 16S rRNA gene amplification and Miseq sequencing and data analysis, the V4–V5 16 S rRNA region were amplified and sequenced using the Illumina MiSeq platform. Duplicate 50-μl PCR reactions were performed, each containing 50 ng of purified DNA, 0.2 mM dNTPs, 1.5 mM MgCl2, 1.25 U Platinum Taq DNA polymerase, 2.5 μl of 10X PCR buffer, and 0.5 μM of each primer designed to amplify the V4-V5: F (5′-ACACTCTTTCCCTACACGACGCTCTTCCGATCTAY TGGGYDTAAAGNG-3′) and R (5′-GTGACTGGAGTTCAGACGTGTGCTCTT CCGATCTCCGTCAATTYHTTTREGT-3′). Cycling conditions were 94 °C for 3 minutes, followed by 22 cycles of 94 °C for 50 seconds, 51 °C for 40 seconds, and 72 °C for 1 minute. In all, 72 °C for 5 min is used for the final elongation step. Amplicons were purified using AxyPrep Mag PCR Clean-up kit (Thermo Fisher Scientific). Up to 15 ng of PCR products were carried forward to library pre-paration using second round PCR. The Illumina primer PCR PE1.0 and index primers were used to allow multiplexing of samples. Eight cycles of enrichment PCR were performed, and final libraries cleaned with AxyPrep Mag PCR Clean-up kit. The library was quantified using ViiA™ 7 Real-Time PCR System (Life Tech-nologies) according to manufacturer's instructions and visualized for size validation on an Agilent 2100 Bioanalyzer (Agilent Technologies) using a high sensitivity DNA assay according to manufacturer's instructions. The sequencing library pool was diluted to 4 nM until run on a MiSeq desktop sequencer (Illumina). In all, 600 cycles chemistry (Illumina) was used according to manufacturer's instructions to run the 6 pM library with 20% PhiX (Illumina), and FASTQ files were used for data analysis. Reads (300 bp paired-end) were merged and then quality-filtered to remove reads with degenerate sites using mothur69, 70 (using the make.contigs and screen.seqs functions, respectively). Reads were then further quality filtered by length (350–375 bp for V4-5). Genus-level assignments per-read were then made using SILVA71 reference sequences in mothur with classify.seqs at 80% confidence. Barplots were created using standard functions in R.

**16S PacBio SMRT Sequencing and data analysis**. For Fig. 3. Microbial DNAs from 1 to 3 mg of murine fecal samples from cecum were extracted and purified according to the manufacturer's protocol of EXT3-16S DNA Purification and PCR Amplification Kit of Shoreline Biome (Farmington, CT), and the extended (~2500 bp) region, which contains 16 S rRNA gene, the adjacent Internally Tran-scribed Spacer and part of the 23 S gene, was amplified using barcoded primer sets in the same kit.

For construction of SMRTbell libraries, all reagents were provided by PacBio (Menlo Park, CA) SMRTbell Template Prep Kit 1.0. Equal molar quantities of the amplicons were pooled, and 50 μl of the DNA repair mixture containing 37 μl of pooled DNA, 5 μl of DNA Damage Repair Buffer (10×), 0.5 μl of NAD+ (100×), 5 μl of ATP high (10 mM), 0.5 μl of dNTP (10 mM), and 2 μl of DNA Damage Repair Mix were incubated at 37 °C for 20 minutes. To generate blunt ends of the DNA, 2.5 μl of End Repair Mix (20×) was treated at 25 °C for 5 minutes. In all, 40 μl of the adapter ligation mixture containing 20 μl of blunted ended DNAs, 5 μl of Annealed Blunt Adapter (20 μM), 4 μl of Template Prep Buffer (10×), 2 μl of ATP low (1 mM) and 1 μl of ligase (30 U/μl) was incubated at 25 °C overnight. To degrade failed ligation products, 1 μl of ExoIII (100.0 U/ul) and ExoVII (10.0 U/ul) was treated to the mixture at 37 °C for 1 hour.

To produce polymerase complexes, all reagents were provided by PacBio (Menlo Park, CA) and the reagent concentrations in the protocol were calculated by the Sample Setup module in PacBio SMRT Link (v8.0.0.80529). The Sequencing Primer v2 (1 μl) was diluted by 29 μl of Elution Buffer and the mixture was incubated at 80 °C for 2 minutes. And, the conditioned sequencing primers were incubated with the library at 20 °C for 1 hour. In all, 1 μl of Sequel Polymerase 3.0 was diluted by 9 μl of Sequel Binding Buffer, first. And, the polymerase was further diluted by adding 1 μl of Sequel Binding Buffer into 3.2 μl of diluted polymerase. The diluted polymerase was applied to the library with the sequencing primers, and the mixture was incubated at 30 °C for 1 hour. After purification of the polymerase complexes using AMPure PB Beads, 85 μl of the final loading dilution in the sample plate was loaded into Sequel. The concentration of the sample on the plate was 8 pM and Movie Time was 10 hours and 2 hours of Pre-Extension Time was applied. The primary analyses, including real-time signal processing and base calling were processed by a built-in PacBio Blade Center through Sequel ICS, and result stream directly to SMRT Link (v8.0.0.80529). The Circular Consensus Sequences (CCS) reads (>5 Minimum Number of Passes and 0.99 Minimum Predicted Accuracy) were produced by the CCS module in SMRT Link (v8.0.0.80529). The demultiplexing and the taxonomic classification analysis of the CCS reads were carried out using SBanalyzer (v2.4-2) of Shoreline Biome (Farmington, CT) based on Athena V2 database.

**Bacteria culture**. Total liver cell suspension was cultured under 5% $CO_2$ in blood agar plates for 24–48 hours at 37 °C.

**Histological analysis**. Tissue specimens were fixed in formalin before embedding in paraffin blocks, sectioned and stained with H&E. Slides were examined at ×100 or ×200 magnification and visualized with a Zeiss Observer II. Each segment of colon was given a score of 0–4: grade 0, no significant changes; grade 1, minimal scattered mucosal inflammatory cell infiltrates, with or without minimal epithelial hyperplasia; grade 2, mild scattered to diffuse inflammatory cell infiltrates, some-times extending into the submucosa and associated with erosions, with mild-to-moderate epithelial hyperplasia and mild to moderate mucin depletion from goblet cells; grade 3, moderate inflammatory cell infiltrates that were sometimes trans-mural, with moderate to severe epithelial hyperplasia and mucin depletion; grade 4, marked inflammatory cell infiltrates that were often transmural and associated with crypt abscesses and occasional ulceration, with marked epithelial hyperplasia, mucin depletion, and loss of intestinal glands. For Paneth cell quantification, a total of nine pictures from three different locations of the H&E-stained slides from one mouse were taken under ×200 magnification, and total Paneth cell and crypt numbers were counted, Paneth cell numbers per crypt are shown.

**Statistics analysis**. Student's unpaired $t$ test was used to compare two groups when data were normally distributed. Mann–Whitney test was used to compare two groups when data were not normally distributed. For comparing more than two groups, Kruskal–Wallis test with Dunn's multiple comparisons test was used was used to compare two groups when data were not normally distributed. One-way analysis of variance (ANOVA) with Tukey's multiple comparisons test, with the Greenhouse–Geisser correction or Student's $t$ test corrected for multiple comparisons using the Holm–Sidak method were used to compare means in paired samples. Ordinary one-way ANOVA with Tukey's correction for multiple com-parisons, two-way ANOVA with Tukey's or Sidak's correction for multiple com-parisons was used to compare means in unpaired samples. Log-rank test was used for survival comparisons. Nonlinear regression (curve fit) was used for bodyweight and diarrhea comparisons. All statistical analyses were performed using GraphPad Prism 8. $P$ values < 0.05 were considered statistically significant.

**Reporting summary**. Further information on research design is available in the Nature Research Reporting Summary linked to this article.

# Data availability
16S PacBio SMRT Sequencing data have been deposited in the GEO database under the accession number GSE159031. 16S Miseq data have been deposited in GEO database under the accession number GSE159418. The two data sets were combined in a superseries under the accession number GSE159419. All other data supporting the findings of this study are available within the article and its supplementary information files or from the corresponding author upon reasonable request. Source data are provided with this paper.

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

## Acknowledgements

This work was supported by National Institutes of Health Grant R01 AI066008 and R01 CA228465 (to D. Zeng), as well as supported by NCI P30CA033572 to City of Hope comprehensive cancer center. We thank Dr. Lieping Chen at Yale University School of Medicine for providing PD-L1$^{-/-}$ BALB/c breeders, Dr. James. Ferrara for providing Reg3$\gamma^{-/-}$ C57BL/6 mice and Genentech for providing anti-mouse IL-22 antibody. We thank Xiwei Wu and his staff at the COH Integrative Genomics Core, Lucy Brown and her staff at the COH Flow Cytometry Facility, Peiguo Chu and his staff at COH Solid

Tumor Core, and Dr. Richard Ermel and his staff at COH Animal Research Center for excellent service support.

## Author contributions

Q.S. designed and performed research as well as prepared the manuscript; D. Zhao provided Reg3γ$^{-/-}$ mice; X. Wang performed experiments. X. Wu., T.H.K., and H.Q. performed 16 S rRNA sequencing and data analysis. R.J. and R.M.V. provided advice on experimental design and critical review and editing manuscript. A.D.R. provide advice and additional financial support for the project as well as critical review of manuscript. P.J.M. provided advice on experimental design and provided critical review and editing of manuscript. Y.C. is the PhD advisor for Q.S. and helped Q.S. on selecting research project and designing experiments. D. Zeng. designed and supervised the research and wrote the manuscript.

## Competing interests

R.R.J. is on the scientific advisory board for Seres Therapeutics, Inc., has consulted for Ziopharm Oncology and MicrobiomeDx, and holds patents licensed to Seres Therapeutics, Inc. The other authors have no conflict of interest.
