## [Peer Review File · Nature Communications]

Reviewers' comments:

Reviewer #1 (GVHD, gut immunity)(Remarks to the Author):

Thank you for giving me the opportunity to review this important study. The manuscript by Song et al is highly interesting as it addresses some very important and timely topics in the field of GVHD especially the mechanisms of gut damage and regeneration. It is also of potential significant clinical relevance as it suggests that IL-22 therapy to promote gut healing after GVHD may be deleterious. It therefore also provides additional insight into the potentially opposing roles of host vs donor derived IL-22 effects on GVHD. Furthermore, the authors attempt to gain a better understanding of the mechanistical underpinnings of SR-GVHD. The authors demonstrate that IL-22 derived from donor Tc17 cells augment host tissue production of Reg3 γ and that and both IL-22 and Reg3 γ are crucial for induction of Tc17-dependent intestinal GVHD and dysbiosis. Depletion of donor CX3CR1+ MNP through PD-L1 is thought to be a consequence of the absence of IFN γ from donor Tc17 cells and contributes to induction of dysbiosis.

Major

1. The most important issue is to address how much this very defined and specific model of CD8/Tc17-dependent GVHD is reflective of - or relevant for the clinical situation of SR-GVHD and predictive of the role of IL-22 in GVHD.
2. Is the deleterious role of IL-22 derived from donor Tc17 cells and the described axis of Reg3 γ /dysbiosis also operative in GVHD models other than B6 to BALB/c (minor mismatch and haplo-mismatch).
3. How much does the conditioning-associated initial tissue damage play a role, i.e. is the Tc17/IL-22 effect also observed in a P to F1 model.
4. Which role does host-derived IL-22 have in this model or this also completely abrogated by the systemic blockade? How do the IFN γ KO/Tc17 cells affect ILC populations in the gut and does IL-22 blockade affect the recipient ILC populations?

Minor:

1. Fig S1 would benefit placing a title label above each of the columns with contour plots.
2. Some typing and spelling errors throughout the manuscript should be corrected.

Reviewer #2 (aGVHD, inflammation)(Remarks to the Author):

The manuscript „Tc17 cells cause gut graft-versus-host disease through induction of dysbiosis and depletion of CX3CR1hi phagocytes,, from Qingxiao Song et al. is a well-structured and complex approach aiming to understand the function of IL22 producing Tc17 cells in the induction of acute gastrointestinal GvHD. These findings displayed through numerous murine models provide a novel insight into the pathophysiology of Tc17 cells in mice. The studies are well performed and the findings are interesting.

The SR GVHD model needs to be explained better. How is refractoriness determined?

Th17 cells are known to contribute to neutrophil recruitment in damaged and inflamed tissues. There are no changes in the Tc17 subsets in the MLN and colon, when mice were transplanted with INF γ -/-

SPL cells and treated with anti-IL22 or with its negative control, however less diarrhea was seen. As neutrophils have a crucial function in the induction of GvHD (Hülsdünker, Blood 2018, Schwab L et al. Nat med 2014), these favorable changes after anti-IL22 treatment might be due to decreased neutrophil recruitment in the colon. Are there any changes in neutrophilic infiltration of recipient gut tissue depending on anti-IL22 treatment?

As described in the manuscript, the literature refers to RegIII γ as a protective antimicrobial endogenous protein, however you observed augmented bacterial translocation in the case of excessive RegIII γ production. Is there an explanation for the controversial function of RegIII γ in the context of INF γ mediated Paneth cell damage?

In patients after allogeneic stem cell transplantation, the administration of broad-spectrum antibiotics is linked to increased GvHD-related mortality (Shono, Sci Transl med 2016), however you found less diarrhea and a prolonged survival of mice. What is the explanation of these conflicting results?

The production of IL22 in mice is largely attributed to Th17 cells, however in human Th1, Th22 and Th17 cells share this function. This information should be provided by the authors in the introduction to enable an adequate interpretation of the data.

Some of the abbreviations are not described at first place leading to difficulties in the understanding.

Reviewer #3 (Gut immunity, microbiota)(Remarks to the Author):

In this manuscript, Song Q., et al, show that upon DEX treatment most T cells get reduced however a sizable population of IL22-producing IL17neg Tc17 appears by day 30. The authors confirmed the relevance of the IL-22 signaling pathway by performing experiments in which IL-22 signaling was blocked. In agreement, ROR γ t-deficient donor T cells failed to induce Gut-GvHD. Since IL-22 signaling induces the antimicrobial peptide Reg3 γ , the authors investigate the contribution of the microbiota in the induction of disease. Classical cohousing and antibiotic experiments demonstrated that the microbiota was required to induce GvHD. The authors showed that CX3CR1 mononuclear cells were affected after IFN-deficient donor CD8 T cells, which was associated to bacterial translocation and induction of GvHD. The authors propose a model in which CX3CR1 MNP produces IL-18BP neutralizing IL-18, which in turn induce the expansion of IL-22 Tc17 cells. IL-22 produced by Tc17 cells act on intestinal epithelial cells to promote Reg3 γ -induced dysbiosis, which in turn promotes GvHD.

Although these findings are potentially interesting, the authors overinterpreted some of the main results. For instance, while the requirement of IL-22 to promote pathology is very convincing, its link with Reg3 γ and dysbiosis is rather weak and counterintuitive. Moreover, the statement that CX3CR1 produce IL-18BP which then trigger the induction of GvHD is not supported by the data. In addition, the role of PD1 and CX3CR1 MNPs in the whole process is somehow confusing. Therefore, some fundamental controls and questions remain to be addressed to support the interpretation of the data in more concrete and accurate conclusions.

Major comments

1. Overall, the manuscript lacks cohesion and fails to provide the rationale for each experiment, which makes it very dense and difficult to be read.
2. The main findings are related to steroid-resistant GvHD, which is not stated in the title. Please include steroid-resistant in the title to avoid misleading generalizations.

3. The authors described a population called IL17-IL22+ Tc17 which increased 25% by day 30 after SR-GvHD. It is not clear why the authors called this population Tc17 if they are IL17neg? Calling them Tc17 without having any feature of IL17-producing cells (e.g. ROR γ t expression) may be misleading. Indeed, IL-22+IL17- CD4 T cells are called Th22 rather than IL-22 producing Th17 cells.
4. Fig S3 indicates that ROR γ t+ T cells are needed to promote pathology. Is this because they are not able to produce IL-22 or due to the lack of Tc17 (IL-17-producing CD8 T cells)? The author should address this point with, for example, rescue experiments using recombinant IL-22 or similar.
5. In line with the previous comment, the text reads: "Furthermore, even in the absence of donor CD4+ T cells, IFN γ -/- CD8+ T. However, cells differentiated into the three subsets: IL-17A+IL-22-, IL-17A+IL-22+ and IL17-IL-22+, and DEX". I assume this was done using IFN-deficient T cells, what if ROR γ t-/- T cells are used?
6. In fig S6 it is shown that IFN γ -deficient donor CD8 T cells result in increased Tc17 differentiation (even IL17+IL-22- T cells). This association becomes somehow confusing.
7. Fig S4A show for the comparison between WT and IFN γ -/- donor splenocytes, in which WT splenocytes do not induce disease. This is an important figure to then understand the use of IFN γ -/- donor cells throughout the paper. The authors should move a panel (like this one) to the main figures.
8. In Fig S9, the authors show RegIII γ expression by immunofluorescence. Since this staining has been controversial in the literature (very few reports showing staining with lack of controls), the authors should test the specificity of their staining by using the RegIII γ -/- mouse.
9. In figure 2F, the authors show that RegIII γ recipients are protected of Gut-GvHD. Is this effect due to an already dysbiotic situation at the time of irradiation and donor cell transfer or is needed after induction of Gut-GvHD?. The authors should perform co-housing experiments prior to induction of Gut-GvHD.
10. The Hooper lab has previously shown that lack of RegIII γ results in increased bacterial association to the epithelium and enhanced activation of the adaptive immune response (Vaishnava S., et al, Science 2011). More recently, a correlation between overproduction of RegIII γ and bacterial depletion at the ileal segment was shown (Jijon H. et al., Muc Immunol, 2018). Therefore, it is somehow counterintuitive that overproduction of RegIII γ would facilitate bacterial translocation to the liver.
11. On page 9 the authors conclude: "These results indicate that IL-22-producing Tc17 cells can change intestinal microbiome profile and cause dysbiosis via enhancing RegIII γ production; in turn, dysbiosis can also augment the expansion of donor Tc17 subsets in recipients of IFN γ -/- SPL cells." However, most of the data provided indicate correlation rather than causality. For example, the authors have not demonstrated that CD8 T cells are the only source of IL-22 required to induce disease. Other IL-22 producers are ILCs and neutrophils in certain scenarios. Moreover, it has not been demonstrated that "IL-22-producing Tc17 cells can change intestinal microbiome profile and cause dysbiosis". The authors should use IL-22-deficient CD8 T cells to demonstrate this point. To demonstrate that overproduction of Reg3 γ result in dysbiosis the authors should analyze the microbiota in Reg3 γ -/- recipients in which GvHD has been induced. Finally; does dysbiosis "cause" the expansion of Tc17 cells? To unequivocally address this, the authors might need to perform colonization of germ-free animals and test Tc17 expansion.
12. In page 15 the authors concluded: "These results indicate that IL-18BP from CX3CR1hi MNP neutralizes IL-18 and reduces expansion of IL-22-producing Tc17 cells in PD-L1-/- recipients." The conclusion that IL18BP is produced by CX3CR1 MNP to trigger a cascade of event that ends in GvHD might be considered misleading. This is due to the lack of data showing that CX3CR1 are the only (or dominant) cell type that produce IL-18BP. It is not demonstrated that actually these cells produced IL-18BP (only mRNA, not protein, analysis). The results shown in Figure 7, in which CX3CR1-KO donor cells are used, do not demonstrate that IL18BP comes from CX3CR1, but that CX3CR1hi cells lead to a higher IL18BP serum concentration (which can be done by a distinct cell type). Therefore, the conclusion by the authors can be considered inaccurate.
13. The conclusion of the PD-L1-/- experiments is that CX3CR1+ cells can affect IL22 production by Tc17 cells. However, the authors never exclude the role of other PD-L1-expressing cell types such as

dendritic cells. CX3CR1+ cell transfer experiments would be needed in order to make this claim.

14. The authors never showed that their depletion antibodies worked. This data or specific transfer of Tc17 cells would be necessary to make the claims in Figures 2-6.

15. In general, the authors should also show absolute cell numbers, gating strategies should be presented and graphs should indicate from which population cell frequencies are derived from (see Figures 1B-C, 2B, 8E, etc).

16. Spleen data is not shown in Figure 1C despite being referred to in the legend.

Minor comments

- In the main text, page 9, it reads: "Higher prevalence of pathogenic Akkermansia, Streptococcus and E. coli", however, Akkermansia has shown tissue regenerative properties, indicating that they are beneficial rather than pathogenic.

- In fig 4A, please indicate in which period (before or during GvHD induction) the mice were co-housed.

- In the abstract, the sentence "IL-22 from Tc17-derived from IFN γ ^{-/-} donor CD8⁺ T cells augments host-tissue production of RegIII γ , leading to dysbiosis." It has some typos and is not possible to understand. Please rewrite it.

- Check typos and consistency in nomenclature.

- Figure S11 lacks legend.

- Page 10 has the title: "Mutual influence of intestinal microbiome and Tc17 cells that produce IL-22." This is an incomplete thought.

Reviewers' comments:

Reviewer #1 (GVHD, gut immunity) (Remarks to the Author):

Major

1. The most important issue is to address how much this very defined and specific model of CD8/Tc17-dependent GVHD is reflective of - or relevant for the clinical situation of SR-GVHD and predictive of the role of IL-22 in GVHD.

From new experiments we learned that the concept "IL-22-producing Tc17 cells" in the previous manuscript was not correct. We have changed "IL-22-producing Tc17 and Tc22 cells" to "Tc22 cells", because most of the IL-22-producing CD8⁺ T cells in the recipients of IFN- γ ^{-/-} CD8⁺ T cells were IL17A⁻IL-22⁺ cells, and they were ROR γ ^t AHR⁺ (Fig.4, and result section on page 10).

Similarly, most of the IL-22-producing CD4⁺ and CD8⁺ T cells in recipients with SR-Gut-aGVHD were IL-17A⁻IL-22⁺ cells that express AHR but not ROR γ ^t (Fig. 2 and result section on page 6-8).

The numbers of IL-22-producing Th/Tc22 cells were expanded in recipients with SR-Gut-aGVHD. IL-22 deficiency in donor T cells and treatment with anti-IL-22 prevented dysbiosis and SR-Gut-aGVHD (Fig.2 and Fig. 3 and result section on pages 6-9). In addition, we induced Xeno-GVHD with PBMC from three healthy donors and found that one of the three Xeno-GVHD became steroid resistant (SR-Gut-aGVHD). We also observed expansion of CD4⁺ and CD8⁺ Th/Tc22 cells in the gut tissue (Fig. S1 and main text on page 7).

2. Is the deleterious role of IL-22 derived from donor Tc17 cells and the described axis of Reg3 γ dysbiosis also operative in GVHD models other than B6 to BALB/c (minor mismatch and haplo-mismatch).

The observation that Tc22 mediate Gut-aGVHD in an IL-22-dependent manner was replicated in a haplo-mismatched HCT model (Fig S7 and result section on page 12).

3. How much does the conditioning-associated initial tissue damage play a role, i.e. is the Tc17/IL-22 effect also observed in a P to F1 model.

TBI-conditioning was required for induction of IL-22-producing Tc22 cells (Fig S7 and result section on page 12).

4. Which role does host-derived IL-22 have in this model or this also completely abrogated by the systemic blockade? How do the IFN γ KO/Tc17 cells affect ILC populations in the gut and does IL-22 blockade affect the recipient ILC populations?

We observed that host-type IL-22-producing ILC3 cells were eliminated before disease onset (Fig. S6) and IL-22-deficiency in donor CD8⁺ T cells prevented induction of Gut-aGVHD by anti-IFN- γ treatment (Fig S5). Thus, IL-22 from host-type innate ILC3 cells are unlikely to contribute to induction or protection of dysbiosis or Gut-aGVHD in recipients given IFN- γ ^{-/-} donor CD8⁺ T cells, and IL-22 blockade is unlikely to affect recipient ILC populations.

Minor:

1. Fig S1 would benefit placing a title label above each of the columns with contour plots.

We moved previous gating strategy in Fig S1 to the Fig S13.

2. Some typing and spelling errors throughout the manuscript should be corrected.

Typographical errors have been corrected throughout the manuscript.

Reviewer #2 (aGVHD, inflammation)(Remarks to the Author):

1. The SR GVHD model needs to be explained better. How is refractoriness determined?

A single injection of DEX at day 3 ameliorated acute GVHD and prevent acute mortality within 10 days, but GVHD recurred by day 20 after HCT. Additional injections of DEX at 10, 15 and 20 days after HCT did not reduce the severity of gut GVHD. Moreover, four injections of DEX (4-DEX) increased serum ST2, Reg3 γ and TNFR1 concentrations,

which have been reported as 3 biomarkers for high-risk Gut-aGVHD. Thus, we consider mice treated with 4-DEX as SR-Gut-aGVHD (Fig. 1 and result section on page 5).

2. Th17 cells are known to contribute to neutrophil recruitment in damaged and inflamed tissues. There are no changes in the Tc17 subsets in the MLN and colon, when mice were transplanted with INF γ ^{-/-} SPL cells and treated with anti-IL22 or with its negative control, however less diarrhea was seen. As neutrophils have a crucial function in the induction of GvHD (Hülsdünker, Blood 2018, Schwab L et al. Nat med 2014), these favorable changes after anti-IL22 treatment might be due to decreased neutrophil recruitment in the colon. Are there any changes in neutrophilic infiltration of recipient gut tissue depending on anti-IL22 treatment?

We observed marked reduction of Ly6G⁺CD11b⁺ neutrophils after neutralizing IL-22 (Fig. 5e and result section page 11). We have cited references (Hülsdünker, Blood 2018, Schwab L et al. Nat med 2014)

2. As described in the manuscript, the literature refers to RegIII γ as a protective antimicrobial endogenous protein, however you observed augmented bacterial translocation in the case of excessive RegIII γ production. Is there an explanation for the controversial function of RegIII γ in the context of IFN γ mediated Paneth cell damage?

RegIII γ is generally protective. Under inflammatory conditions, RegIII γ can reduce diversity of intestinal microbiome and augment pathogen colonization (Bensen et al: Immunity 2014, now cited on page 8). In addition, we observed that loss of CX3CR1^{hi} MNP in recipients given IFN- γ ^{-/-} CD8⁺ T or SR-Gut-GVHD recipients also contribute to enhanced bacterial translocation (Fig. 9 & 10 and result section on pages 17-18).

3. In patients after allogeneic stem cell transplantation, the administration of broad-spectrum antibiotics is linked to increased GvHD-related mortality (Shono, Sci Transl med 2016), however you found less diarrhea and a prolonged survival of mice. What is the explanation of these conflicting results?

Broad-spectrum antibiotics cause dysbiosis in patients and lead to worsening of gut GVHD (Shono et al: Sci Transl med 2016). When we used combination of 4 antibiotics to eliminate the bacteria in the feces from ileum on day 6, as indicated by undetectable 16S, we also eliminated dysbiosis and prevented induction of Gut-aGVHD by IFN- γ ^{-/-} CD8⁺ T cells. These results indicate that dysbiosis induced by Tc22-derived from IFN- γ ^{-/-} CD8⁺ T cells has a critical pathogenic role (Fig.7 and result section page 14-15).

4. The production of IL22 in mice is largely attributed to Th17 cells, however in human Th1, Th22 and Th17 cells share this function. This information should be provided by the authors in the introduction to enable an adequate interpretation of the data.

We have added this information to the introduction (page 3).

5. Some of the abbreviations are not described at first place leading to difficulties in the understanding.

We have spelled out all abbreviations throughout the text.

Reviewer #3 (Gut immunity, microbiota) (Remarks to the Author):

Major comments

1. Overall, the manuscript lacks cohesion and fails to provide the rationale for each experiment, which makes it very dense and difficult to be read.

The text has been extensively edited to ensure clarity.

2. The main findings are related to steroid-resistant GvHD, which is not stated in the title. Please include steroid-resistant in the title to avoid misleading generalizations.

We agree and have changed the title to “Steroid-resistant gut graft-versus-host disease results from expansion of Th/Tc22 cells, dysbiosis, and depletion of CX3CR1^{hi} phagocytes”.

3. The authors described a population called IL17⁻IL22⁺ Tc17 which increased to 25% by day 30 after SR-GvHD. It is not clear why the authors called this population Tc17 if they are IL17^{neg}? Calling them Tc17 without having any feature of IL17-producing cells (e.g. ROR γ t expression) may be misleading. Indeed, IL-22⁺IL17⁻ CD4 T cells are called Th22 rather than IL-22 producing Th17 cells.

We agree that they are Th/Tc22 cells, because our new data showed that the IL-22⁺IL-17⁻ CD4⁺ and CD8⁺ T cells are AHR⁺ and ROR γ t⁻ (Fig.2d and Fig4f).

4. Fig S3 indicates that ROR γ t⁺ T cells are needed to promote pathology. Is this because they are not able to produce IL-22 or due to the lack of Tc17 (IL-17-producing CD8 T cells)? The author should address this point with, for example, rescue experiments using recombinant IL-22 or similar.

Due to word limitations in the main text, the context related to previous Fig. S3 has been deleted.

5. In line with the previous comment, the text reads: “Furthermore, even in the absence of donor CD4+ T cells, IFN γ ^{-/-} CD8+ T. However, cells differentiated into the three subsets: IL-17A+IL-22⁻, IL-17A+IL-22⁺ and IL17-IL-22⁺, and DEX”. I assume this was done using IFN-deficient T cells, what if ROR γ t^{-/-} T cells are used?

Both IFN- γ ^{-/-} and IFN- γ ^{-/-}ROR γ t^{-/-} T cells gave rise to IL17⁺IL-22⁺ Tc22 subsets (Fig.4).

6. In fig S6 it is shown that IFN γ -deficient donor CD8 T cells result in increased Tc17 differentiation (even IL17+IL-22⁻ T cells). This association becomes somehow confusing.

Previous Fig S6 has become Fig. 4 after revision. As mentioned above, our previous description of IL17⁺IL-22⁻ T cells as Tc17 was wrong, and we now describe them as Tc22 cells.

7. Fig S4A show for the comparison between WT and IFN γ ^{-/-} donor splenocytes, in which WT splenocytes do not induce disease. This is an important figure to then understand the use of IFN γ ^{-/-} donor cells throughout the paper. The authors should move a panel (like this one) to the main figures.

This supplemental data has been moved to the main text in Fig.4.

8. In Fig S9, the authors show RegIII γ expression by immunofluorescence. Since this staining has been controversial in the literature (very few reports showing staining with lack of controls), the authors should test the specificity of their staining by using the RegIII γ ^{-/-} mouse.

Due to concerns about the lack of RegIII γ ^{-/-} controls, we have deleted these results in the revised manuscript.

9. In figure 2F, the authors show that RegIII / recipients are protected of Gut-GvHD. Is this effect due to an already dysbiotic situation at the time of irradiation and donor cell transfer or is needed after induction of Gut-GvHD? The authors should perform co-housing experiments prior to induction of Gut-GvHD.

We understand the reviewer’s concern. Although both WT and RegIII γ ^{-/-} mice appear to be healthy, RegIII γ ^{-/-} mice have low-levels of dysbiosis (Loonen LM, et al. Mucosal immunology 7, 939-947, 2014). In our experiments, RegIII γ is pathogenic rather than protective, and RegIII γ deficient recipients had lower prevalence of pathogenic E. Coli and higher prevalence of protective Barnesiella and Blautia, with less severe Gut-aGVHD. Although cohousing the WT and RegIII γ ^{-/-} mice before HCT would be ideal, given the pathogenic role of RegIII γ in our experiments, we do not think it is necessary to repeat these experiments with pre-co-housed WT and RegIII γ ^{-/-} mice.

10. The Hooper lab has previously shown that lack of RegIII γ results in increased bacterial association to the epithelium and enhanced activation of the adaptive immune response (Vaishnava S., et al, Science 2011). More recently, a correlation between overproduction of RegIII γ and bacterial depletion at the ileal segment was shown (Jijon H. et al., Muc Immunol, 2018). Therefore, it is somehow counterintuitive that overproduction of RegIII γ would facilitate bacterial translocation to the liver.

We understand the reviewer's concern. Although over-production of IL-22 and RegIII γ caused dysbiosis (Fig. 3 and 6), dysbiosis alone did not induce bacterial translocation and additional depletion of CX3CR1^{hi} MNP was required to augment bacterial translocation (Fig.9). We have stressed this point in discussion (page20-21).

11. On page 9 the authors conclude: "These results indicate that IL-22-producing Tc17 cells can change intestinal microbiome profile and cause dysbiosis via enhancing RegIII γ production; in turn, dysbiosis can also augment the expansion of donor Tc17 subsets in recipients of IFN γ -/- SPL cells." However, most of the data provided indicate correlation rather than causality. For example, the authors have not demonstrated that CD8 T cells are the only source of IL-22 required to induce disease. Other IL-22 producers are ILCs and neutrophils in certain scenarios. Moreover, it has not been demonstrated that "IL-22-producing Tc17 cells can change intestinal microbiome profile and cause dysbiosis". The authors should use IL-22-deficient CD8 T cells to demonstrate this point. To demonstrate that overproduction of RegIII γ result in dysbiosis the authors should analyze the microbiota in RegIII γ -/- recipients in which GvHD has been induced. Finally; does dysbiosis "cause" the expansion of Tc17 cells? To unequivocally address this, the authors might need to perform colonization of germ-free animals and test Tc17 expansion.

New data show that IL-22 deficiency in CD8⁺ T cells prevented induction of gut-aGVHD and prevented bacterial translocation (Fig.S5 and Results on page 11-12). We also showed that RegIII γ deficiency in recipients prevented dysbiosis, Gut-aGVHD and bacterial translocation induced by IFN- γ ^{-/-} CD8⁺ T cells (Fig.5 & Fig. 6). We have removed the statement that dysbiosis causes expansion of Tc17 cells and the related data to correct overstatement.

12. In page 15 the authors concluded: "These results indicate that IL-18BP from CX3CR1^{hi} MNP neutralizes IL-18 and reduces expansion of IL-22-producing Tc17 cells in PD-L1-/- recipients." The conclusion that IL18BP is produced by CX3CR1 MNP to trigger a cascade of event that ends in GvHD might be considered misleading. This is due to the lack of data showing that CX3CR1 are the only (or dominant) cell type that produce IL-18BP. It is not demonstrated that actually these cells produced IL-18BP (only mRNA, not protein, analysis). The results shown in Figure 7, in which CX3CR1-

KO donor cells are used, do not demonstrate that IL18BP comes from CX3CR1, but that CX3CR1^{hi} cells lead to a higher IL18BP serum concentration (which can be done by a distinct cell type). Therefore, the conclusion by the authors can be considered inaccurate.

We understand the reviewer's concern and have removed the data and related text.

13. The conclusion of the PD-L1^{-/-} experiments is that CX3CR1⁺ cells can affect IL22 production by Tc17 cells. However, the authors never exclude the role of other PD-L1-expressing cell types such as dendritic cells. CX3CR1⁺ cell transfer experiments would be needed in order to make this claim.

Besides PD-L1^{-/-} experiments (Fig 9b), we observed Tc22 expansion in recipients of CX3CR1^{-/-} donor bone marrow cells (Fig 9f) supporting the conclusion that depletion of CX3CR1^{hi} MNP leads to expansion of Tc22. New transfer experiments with CX3CR1^{hi} and CX3CR1^{lo} MNP showed donor-type CX3CR1^{hi} but not CX3CR1^{lo} MNP ameliorated Gut-aGVHD (Fig. 9h and result section page 17).

14. The authors never showed that their depletion antibodies worked. This data or specific transfer of Tc17 cells would be necessary to make the claims in Figures 2-6.

The depleting anti-CD4 mAb is used in our recent JCI publication showing that injection of anti-CD4 mAb deplete CD4⁺ T cells (Song et al: JCI 2017).

15. In general, the authors should also show absolute cell numbers, gating strategies should be presented and graphs should indicate from which population cell frequencies are derived from (see Figures 1B-C, 2B, 8E, etc).

The gating strategy is shown in (Fig.S13).

16. Spleen data is not shown in Figure 1C despite being referred to in the legend.

The error has been corrected.

Minor comments

- In the main text, page 9, it reads: "Higher prevalence of pathogenic Akkermansia, Streptococcus and E. coli", however, Akkermansia has shown tissue regenerative properties, indicating that they are beneficial rather than pathogenic.

We have removed data regarding Akkermansia and corrected the overstatement.

- In fig 4A, please indicate in which period (before or during GvHD induction) the mice were co-housed.

The mice were co-housed during GVHD induction (See Fig 7 in the revised manuscript).

- In the abstract, the sentence “IL-22 from Tc17-derived from IFN γ -/- donor CD8+ T cells augments host-tissue production of RegIII γ , leading to dysbiosis.” It has some typos and is not possible to understand. Please rewrite it.

The text has been corrected.

- Check typos and consistency in nomenclature.

The text has been corrected.

- Figure S11 lacks legend.

The figure legend has been added.

- Page 10 has the title: “Mutual influence of intestinal microbiome and Tc17 cells that produce IL-22.” This is an incomplete thought.

This statement has been removed.

REVIEWERS' COMMENTS

Reviewer #2 (Remarks to the Author):

The authors have well responded to my comments.

Reviewer #3 (Remarks to the Author):

The authors have satisfactorily addressed all my concerns. A minor concern would be:

- In figure 7b, the authors showed % of clostridiaceae, which seems to be significant between non-cohoused and cohoused mice, however it is only one out of 5-6 samples that resulted with increased abundance while the rest were basically closed to zero. The authors do not refer to this data in the text, therefore it is not clear what the data means. Please clarify or remove

We have addressed the concern from Review 3 regarding Figure 7b (see below).

Reviewer #3 (Remarks to the Author):

The authors have satisfactorily addressed all my concerns. A minor concern would be:

- In figure 7b, the authors showed % of clostridiaceae, which seems to be significant between non-cohoused and cohoused mice, however, it is only one out of 5-6 samples that resulted with increased abundance while the rest were basically closed to zero. The authors do not refer to this data in the text, therefore it is not clear what the data means. Please clarify or remove.

Response: We have modified the Y axis of Figure 7b to make sure that the differences between the two groups are visible. We also mentioned the difference in the main text (see page 15).